*The Company of*
**Biologists**

## RESEARCH ARTICLE

# Metabolic stress and muscle mechanics: Acute response of isolated soleus and EDL muscles to prolonged fasting in mice with distinct muscle phenotypes

Leonardo Cesanelli[1,*,‡], Berta Ylaite[1,*], Marius Brazaitis[1], Nerijus Eimantas[1], Aivaras Ratkevicius[1,2], Danguole Satkunskiene[3] and Petras Minderis[1]

## ABSTRACT

Prolonged fasting impacts skeletal muscle by inducing atrophy, thereby limiting contractile capacity and altering tissue mechanical behavior. This study investigated the effects of 48 h of fasting (FAS) versus *ad libitum* food consumption (CON) on the mechanical properties of fast-twitch (extensor digitorum longus, EDL) and slow-twitch (soleus, SOL) muscles in three mouse strains with distinct muscle phenotypes: C57BL/6J (normal-sized), BEH+/+ (larger muscles), and BEH (myostatin-deficient with markedly larger muscles). Isolated SOL and EDL were subjected to 100 isometric–eccentric contraction cycles, and peak and specific force, rate of force development, fatigue, stiffness, and tangent modulus were assessed. Fasting significantly reduced muscle size and force production capacity (isometric and eccentric) across all strains ($P<0.05$). SOL muscles showed a greater decline in tetanic force (fatigue index: SOL 67% versus EDL 33%, $P<0.05$), while BEH mice exhibited the steepest contractile impairment ($P<0.05$). Fasting also reduced stiffness and tangent modulus across all strains and muscle types ($P<0.05$). These findings demonstrate that fasting consistently impairs contractile and mechanical properties of skeletal muscle, with slow-twitch muscles and larger muscles phenotypes being particularly vulnerable. Muscle type and genetic background thus play key roles in determining the extent of functional decline under metabolic stress.

KEY WORDS: Fasting, Muscle atrophy, Contractility, Mechanobiology, Muscle mechanics

## INTRODUCTION

Fasting, in its various forms, has become a widely adopted practice, with applications ranging from health optimization to clinical interventions (Longo and Mattson, 2014; Jaspers et al., 2017). In the general population, intermittent fasting (e.g. time-restricted feeding) has gained popularity for weight management and metabolic health, with studies suggesting improvements in insulin sensitivity, lipid profiles, and even longevity (Longo et al., 2021). In clinical settings, chronic caloric restriction (i.e. reduced but sustained food intake), but also transient or complete fasting (i.e. total absence of food intake), are often employed to manage different clinical conditions (Wilhelmi de Toledo et al., 2020). For instance, prolonged fasting (e.g. 6–8 days) has recently been the focus of human trials investigating its effects on metabolism, insulin secretion, physical performance, and oxidative stress (Solianik et al., 2023; Brazaitis et al., 2025; Kolnes et al., 2025; Pilis et al., 2025). In addition, fasting is increasingly being explored as a potential strategy for enhancing physical performance, particularly in endurance sports and weight management contexts, with contrasting results (Levy and Chu, 2019; Ashtary-Larky et al., 2021; Zouhal et al., 2020; Johnstone, 2015). Despite the lack of clear evidence supporting these practices (Zouhal et al., 2020; Keenan et al., 2022), in the context of exercise and physical performance, examples include endurance athletes who occasionally train in a fasted state to improve lipid metabolism and optimize performance, and bodybuilders who use fasting to refine body composition, particularly in preparation for competitions. Regardless of its widespread use, the effects of fasting –particularly prolonged fasting – on the structural and functional properties of skeletal muscle remain poorly understood.

Fasting induces significant metabolic shifts, including altered energy substrate availability and stress/inflammatory responses, which can influence the mechanical properties of musculotendinous tissues (Lapinskas et al., 2020; Fokin et al., 2019; Schmidt et al., 2020; Pedroso et al., 2020). While acute fasting may transiently increase inflammation due to oxidative stress and lipid mobilization, prolonged fasting can partially attenuate it by reducing pro-inflammatory cytokines (e.g. TNF-α, IL-6) and promoting a metabolic shift toward ketone utilization. Fasting-induced activation of autophagy and proteolysis may compromise tissue integrity, while also facilitating the clearance of damaged proteins (Longo and Mattson, 2014). These opposing effects make it unclear how fasting ultimately influences contractile performance, recovery, and susceptibility to injury.

Prolonged fasting also affects skeletal muscle contractility by limiting energy supply and disrupting key molecular processes involved in force production (Lapinskas et al., 2020; Schmidt et al., 2020; Bak et al., 2018). As glycogen depletion and proteolysis progress (Lapinskas et al., 2020; Schmidt et al., 2020), actin– myosin interactions that govern cross-bridge cycling and active tension generation may be impaired due to ATP depletion and contractile protein degradation. This energy deficit can hinder cross-bridge formation and detachment, reducing both force and

[1]Institute of Sport Science and Innovations, Lithuanian Sports University, Kaunas 44221, Lithuania. [2]Sports and Exercise Medicine Centre, Queen Mary University of London, London E1 4NS, UK. [3]Department of Health Promotion and Rehabilitation, Lithuanian Sports University, Kaunas 44221, Lithuania.
*These authors contributed equally to the present work and are designated as co-first authors.

‡Author for correspondence (leonardo.cesanelli@lsu.lt)

L.C., 0000-0003-2822-1836; B.Y., 0000-0002-1930-2012; M.B., 0000-0003-1369-7524; N.E., 0000-0003-0955-2090; A.R., 0000-0002-4737-5817; D.S., 0000-0002-2008-6150; P.M., 0000-0001-9522-6050

efficiency. Schmidt et al. (2020) highlighted fasting-induced declines in contractile function under hypoxic conditions, emphasizing how energy depletion and environmental stress may interact to impair muscle performance.

Although a few studies have explored this aspect, fasting may also affect also the biomechanical properties of muscles, by altering their structural and molecular components. Metabolic stress and prolonged physical inactivity – a potential consequence of stressful conditions such as low energy availability – may trigger extracellular matrix (ECM) remodeling (Naba, 2024; Binder-Markey et al., 2021; DeVallance et al., 2017). At the cellular level, costameric proteins, which anchor sarcomeres to the sarcolemma and facilitate lateral force transmission, rely on adequate energy supply for proper assembly and function; insufficient energy could therefore indirectly impair their function and the mechanical integrity of the muscle (Anastasi et al., 2008; Garcia-Pelagio and Bloch, 2021). Similarly, sarcomeric proteins such as titin, that modulates stiffness and elastic recoil, could be compromised by post-translational modifications such as phosphorylation-dephosphorylation, which may be disrupted due to fasting-related ATP depletion and inflammatory stress (Russell and Solís, 2021; Wang et al., 2023).

The time course of such changes may be phenotype dependent. Extensor digitorum longus (EDL; fast-twitch) and soleus (SOL; slow-twitch) muscles differ in their structural components (James et al., 1995), contractile behavior (Mendias et al., 2006), and mechanical properties (Cesanelli et al., 2024; Tyganov et al., 2023). These differences can be further modulated by genotype-specific traits, such as variations in muscle size across mouse strains (Lapinskas et al., 2020; Fokin et al., 2019; Mendias et al., 2006; Satkunskiene et al., 2019). Moreover, EDL and SOL muscles of mice with dysfunctional myostatin were less resilient to repeated contraction-stretch loading, with induced damage leading to greater losses in isometric force production and stiffness than in functional myostatin counterpart mice, particularly in EDL (Satkunskiene et al., 2019). This suggests that even under physiological conditions (i.e. adequate energy availability), myostatin influences muscle mechanical properties in a strain- and muscle-specific manner (Cesanelli et al., 2024; Satkunskiene et al., 2019). Fasting may further exacerbate these phenomena through accelerated proteolysis, glycogen depletion, and altered extracellular matrix remodeling. While the metabolic response to fasting may differ between fiber types – with slow-twitch muscles relying more on lipid oxidation and fast-twitch muscles being

more sensitive to glycogen depletion – SOL muscles may still experience vulnerability due to structural features, pre-existing myostatin dysfunction, and higher metabolic demands associated with larger muscle size. Collagen degradation and ECM remodeling could further impair mechanical integrity and force transmission, amplifying susceptibility to fasting-induced atrophy and contractile deficits.

Therefore, we aimed to investigate the effects of 48 h of fasting on the mechanical properties of two distinct skeletal muscles – the fast-twitch EDL and slow-twitch SOL – in genetically distinct mouse models representing diverse muscle phenotypes: classic C57BL6 strain (normal-sized muscles), BEH+/+ strain (characterized by larger muscles), and BEH strain (Berlin High; non-functioning myostatin mice with abnormally oversized muscles).

## RESULTS

A total of 96 isolated muscles were tested, of which 61 (−35%) successfully completed the 100 contraction-stretching cycles. Most tissue failures (i.e. rupture/damage preventing continuation of the protocol, not slippage or technical error) occurred between the 1st and 10th contraction (−30%) (Table 1). Significant muscle type and strain effects were observed, with EDL being more prone to failure than SOL ($P<0.021$), and C57BL/6J tissues more resilient than BEH+/+ and BEH (failure rates: −25%, −37%, and −52%, respectively; $P<0.037$). Fasting had no significant effect on tissue failure (control group, CON: −38%; fasting group, FAS: −40%).

### Body mass and muscle size

Fasting had a significant overall effect in reducing body mass ($M_B$), muscle mass ($M_M$) and cross-sectional area (CSA) ($P<0.002$, $\eta^2=0.066$) (Table 2). However, similar effects of fasting were observed on both SOL and EDL muscle size and this effect did not differ across the three strains, with no significant condition x muscle type and condition x strain interaction observed ($P>0.05$) (Table 2). The passive and active force–length curves, from which the muscle $L_0$ across conditions and genotypes were determined, are shown in Fig. S1.

### Twitch and force-frequency properties

Twitch force was significantly influenced by genotype ($P<0.001$, $\eta^2=0.863$), condition ($P<0.001$, $\eta^2=0.486$), and muscle type ($P<0.001$, $\eta^2=0.864$), with BEH mice showing the highest

**Table 1. Descriptive data (absolute value and % difference [failure rate]) for SOL and EDL muscles able to successfully complete the protocol (failure rate), across the different stages, in CON and FAS groups across the three mouse strains (C57BL/6J, BEH, and BEH+/+)**

| | | | 1st contraction | 10th contraction | | 50th contraction | | 100th contraction | |
|---|---|---|---|---|---|---|---|---|---|
| | | | n | n | % | n | % | n | % |
| C57BL/6J | CON | n EDL | 6 | 4 | −33 | 4 | −33 | 4 | −33 |
| | | n SOL | 8 | 7 | −12 | 7 | −12 | 7 | −12 |
| | FAS | n EDL | 7 | 4 | −43 | 4 | −43 | 4 | −43 |
| | | n SOL | 9 | 8 | −11 | 8 | −11 | 8 | −11 |
| BEH+/+ | CON | n EDL | 10 | 5 | −50 | 5 | −50 | 5 | −50 |
| | | n SOL | 10 | 6 | −40 | 6 | −40 | 6 | −40 |
| | FAS | n EDL | 10 | 6 | −40 | 6 | −40 | 6 | −40 |
| | | n SOL | 10 | 9 | −10 | 8 | −20 | 8 | −20 |
| BEH | CON | n EDL | 4 | 3 | −25 | 3 | −25 | 2 | −50 |
| | | n SOL | 7 | 4 | −43 | 4 | −43 | 4 | −43 |
| | FAS | n EDL | 6 | 5 | −17 | 3 | −50 | 1 | −84 |
| | | n SOL | 9 | 8 | −11 | 6 | −33 | 6 | −33 |

EDL, extensor digitorum longus; SOL, soleus; CON, control condition; FAS, fasting condition; n, number of isolated tissues; %, percentage of reduction of tissues able to complete the protocol.

**Table 2. Descriptive data (mean±s.d.) for SOL and EDL muscle size parameters (optimal length [L₀], muscle mass [Mₘ], and cross-sectional area [CSA]) in CON and FAS groups across the three mouse strains (C57BL/6J, BEH, and BEH+/+)**

| | C57BL/6J | | | |
| --- | --- | --- | --- | --- |
| | CON | | FAS | |
| $M_B$ 0-h (g) | 26.6±1.45 | | 26.7±1.73 | |
| $M_B$ 48-h (g) | 26.2±1.26 | | 20.2±1.45$^\theta$ | |
| | **EDL** | | **SOL** | |
| | CON | FAS | CON | FAS |
| $M_M / M_B$ ratio | 0.38±0.02 | 0.45±0.02$^\theta$ | 0.31±0.03 | 0.37±0.02$^\theta$ |
| $L_0$ (mm) | 14.6±0.84 | 14.9±0.30 | 13.4±0.53 | 13.5±0.57 |
| $M_M$ (mg) | 12.3±0.52 | 10.9±0.70$^\theta$ | 9.91±0.69 | 8.88±0.88$^\theta$ |
| CSA (mm²) | 1.73±0.12 | 1.51±0.11$^\theta$ | 1.00±0.07 | 0.89±0.09$^\theta$ |
| | **BEH+/+** | | | |
| | CON | | FAS | |
| $M_B$ 0-h (g) | 48.2±1.83 | | 48.1±2.28 | |
| $M_B$ 48-h (g) | 47.3±1.94 | | 36.5±2.04$^\theta$ | |
| | **EDL** | | **SOL** | |
| | CON | FAS | CON | FAS |
| $M_M / M_B$ ratio | 0.27±0.01 | 0.32±0.02$^\theta$ | 0.21±0.01 | 0.27±0.02$^\theta$ |
| $L_0$ (mm) | 16.0±0.85 | 15.5±0.73 | 14.9±0.41 | 14.8±0.97 |
| $M_M$ (mg) | 15.6±1.02 | 15.3±0.86$^\theta$ | 12.9±0.95 | 12.5±1.09$^\theta$ |
| CSA (mm²) | 2.00±0.18 | 2.03±0.08$^\theta$ | 1.16±0.09 | 1.13±0.07$^\theta$ |
| | **BEH** | | | |
| | CON | | FAS | |
| $M_B$ 0-h (g) | 56.2±1.96 | | 56.7±3.19 | |
| $M_B$ 48-h (g) | 56.4±1.56 | | 43.9±2.38$^\theta$ | |
| | **EDL** | | **SOL** | |
| | CON | FAS | CON | FAS |
| $M_M / M_B$ ratio | 0.49±0.01 | 0.56±0.03$^\theta$ | 0.31±0.01 | 0.38±0.01$^\theta$ |
| $L_0$ (mm) | 17.6±0.57 | 18.3±0.60 | 15.6±0.24 | 15.3±0.19 |
| $M_M$ (mg) | 33.5±1.87 | 30.9±1.98$^\theta$ | 21.4±1.41 | 20.5±0.89$^\theta$ |
| CSA (mm²) | 3.90±0.15 | 3.48±0.33$^\theta$ | 1.85±0.12 | 1.80±0.08$^\theta$ |

EDL, extensor digitorum longus; SOL, soleus; CON, control condition; FAS, fasting condition; $M_B$, body mass; $L_0$, optimal length; $M_M$, muscle mass; CSA, cross-sectional area; θ, condition effect $P<0.002$.

absolute twitch force, followed by BEH+/+ and B6. Fasting significantly reduced twitch force across all genotypes and both muscle types, with the SOL being consistently weaker than EDL (Fig. 2B,G). Specific twitch force was also affected by genotype ($P<0.001$, $\eta^2=0.185$) and condition ($P<0.001$, $\eta^2=0.330$), with fasting-induced reductions in stress most pronounced in BEH and BEH+/+, while the muscle type effect was smaller and not significant ($P=0.127$, $\eta^2=0.022$) (Fig. 2C,H). Contraction time

(CT) was significantly longer in SOL compared to EDL ($P<0.001$, $\eta^2=0.863$) and varied across genotypes ($P<0.001$, $\eta^2=0.227$) (Fig. 2D,I). Similarly, half-relaxation time (HRT) was longer in SOL ($P<0.001$, $\eta^2=0.686$), but neither genotype ($P=0.100$, $\eta^2=0.043$) nor condition ($P=0.325$, $\eta^2=0.009$) had significant effects, although there was a trend for reduced HRT in fasted BEH mice (Fig. 2E,J).

Force-frequency relationship was significantly influenced by genotype ($P<0.001$, $\eta^2=0.556$), condition ($P<0.001$, $\eta^2=0.445$), and muscle type ($P<0.001$, $\eta^2=0.752$), with notable interactions between genotype and muscle ($P<0.001$, $\eta^2=0.384$), and between condition and muscle ($P=0.002$, $\eta^2=0.093$). BEH mice generated the highest absolute tetanic forces, followed by BEH+/+ and B6, but also exhibited the most pronounced reduction in force under fasting conditions, especially in the SOL. SOL muscles were more severely affected by fasting across all genotypes (Fig. 3).

## Isometric and eccentric mechanical properties

For isometric contractile function, a significant Condition×Strain interaction ($P<0.034$) indicated that the effect of fasting on isometric contractility varied among mouse strains, with BEH being most affected, followed by BEH+/+ and C57BL/6J (Figs 4 and 5). Similarly, a significant Condition×Muscle Type interaction ($P=0.006$) revealed that the impact of fasting was more pronounced in SOL than EDL. A Strain×Muscle Type×Time interaction ($P<0.001$) further highlighted muscle- and strain-specific fatigue responses over time, with BEH SOL showing the steepest decline (Figs 4 and 5). Overall, fasting significantly reduced isometric force production, rate of force development, and RFD/PF ratio across all strains and muscle types (condition effect: $P<0.001$), with the FAS group already showing lower values at the first contraction ($P<0.01$) (Table S1). These reductions persisted through the 100th contractions, with significant time effects ($P<0.001$). A main effect of muscle type was found for rate of force development and RFD/PF ($P<0.001$), with SOL experiencing greater losses than EDL. BEH strain showed significantly greater isometric contractile capacity compared to BEH+/+ and C57BL/6J ($P<0.001$) but also demonstrated the greatest decline in isometric force over time, as reflected in the lowest fatigue index (FI%) ($P<0.001$) (Tables S1 and S3). For eccentric contractile function, no significant Condition×Strain or Condition×Muscle Type interactions were observed. Fasting significantly reduced peak eccentric force and force enhancement across all mouse strains ($P<0.001$), with the greatest reductions observed in BEH, followed by BEH+/+ and

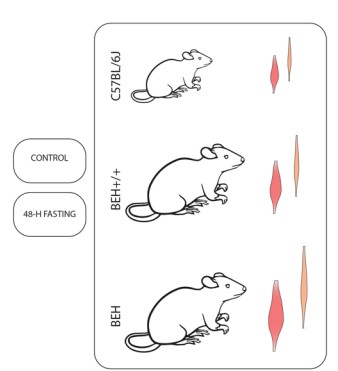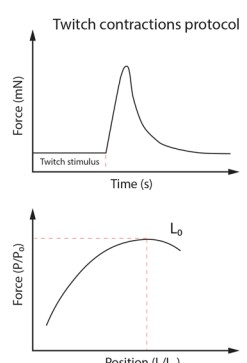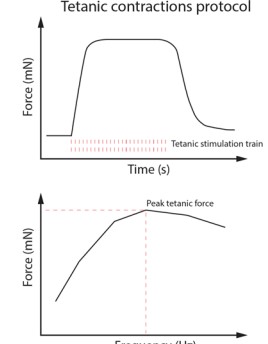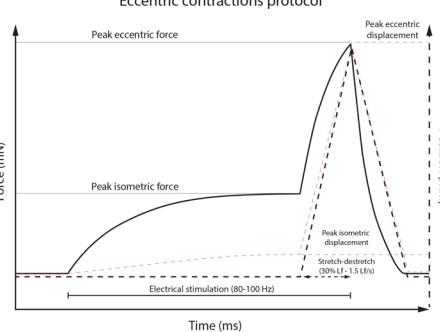

**Fig. 1. Study design and muscle mechanics testing protocols.** Mice from three different strains (C57BL/6J, BEH+/+, and BEH) were randomly assigned to either a control (CON) or 48-h fasting (FAS) group. Isolated SOL and EDL muscles were used for mechanical testing, which included including twitch contractions and determination of optimal length (L0), tetanic contractions at different stimulation frequencies with force–frequency curve analysis, and an isometric–eccentric protocol repeated for 100 cycles.

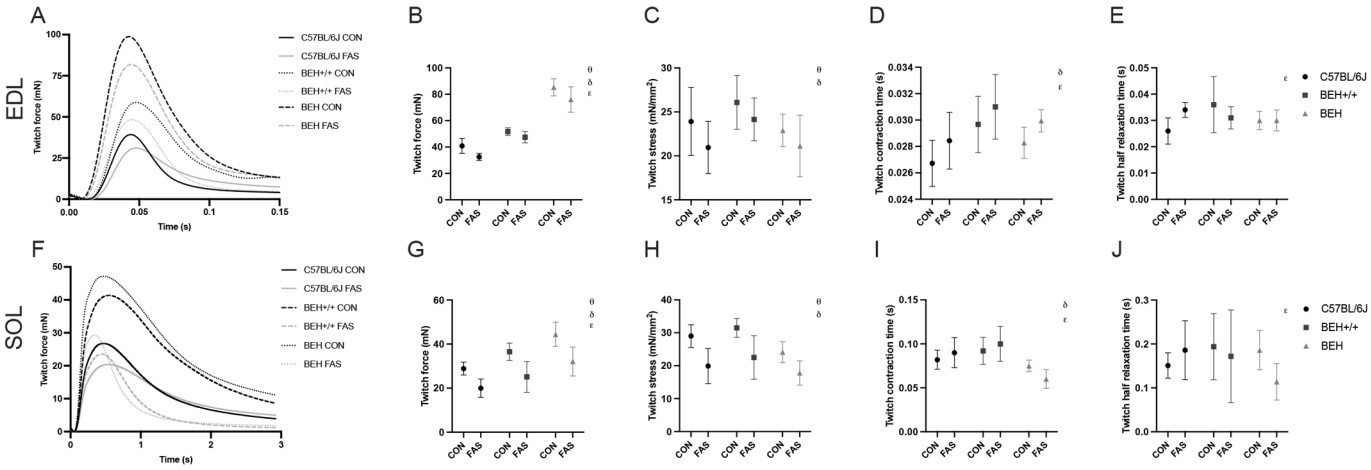

**Fig. 2. Twitch contraction parameters in EDL and SOL muscles across genotype and fasting conditions.** (A,F) example of twitch force-time curves, (B,G) peak twitch force (mN), (C,H) specific twitch force (mN/mm²), (D,I) contraction time (CT), and (E,J) half-relaxation time (HRT). Data are shown as mean ±s.d. θ: significant condition effect, $P<0.05$; δ: significant strain effect, $P<0.05$; ε: significant muscle type effect, $P<0.05$.

C57BL/6J. Fasting also led to significantly greater eccentric force loss in SOL compared to EDL ($P<0.042$). Both eccentric force and force enhancement declined progressively across time points ($P<0.001$), with a similar time course between groups (Fig. 5). The eccentric biomechanical properties (stiffness and tangent modulus) revealed a significant condition effect ($P<0.001$), with fasting reducing all measured properties (Fig. 4). Time effects were significant ($P<0.001$), showing a progressive decline in stiffness and modulus over the course of eccentric contractions (Fig. 4 and Table S2). Strain-specific differences emerged, with BEH showing higher stiffness but lower tangent modulus than BEH+/+ and C57BL/6J at baseline and also exhibiting greater declines after fasting ($P<0.03$). At baseline, significant muscle type effect was also observed for TM$_{ECC}$, with SOL characterized by higher values than EDL ($P<0.03$).

## DISCUSSION

In this study, we investigated the impact of 48-h fasting on muscle mechanics, including contractile function and biomechanical properties, in slow-twitch (SOL) and fast-twitch (EDL) muscles across three distinct phenotypes in term of muscle size (C57BL/6J, BEH+/+, BEH). Our findings demonstrate that (I) fasting significantly reduced contractile function, both isometric and eccentric, across all strains, with SOL muscles showing the greatest decline, and myostatin dysfunctional BEH mice exhibiting the largest loss in contractility; (II) fasting impaired biomechanical properties, such as stiffness, and tangent modulus, consistently across all strains and muscle types.

### Muscle size

Our results showed that fasting reduced muscle size parameters (e.g. TM and CSA), with no specific effect on either SOL or EDL, nor on

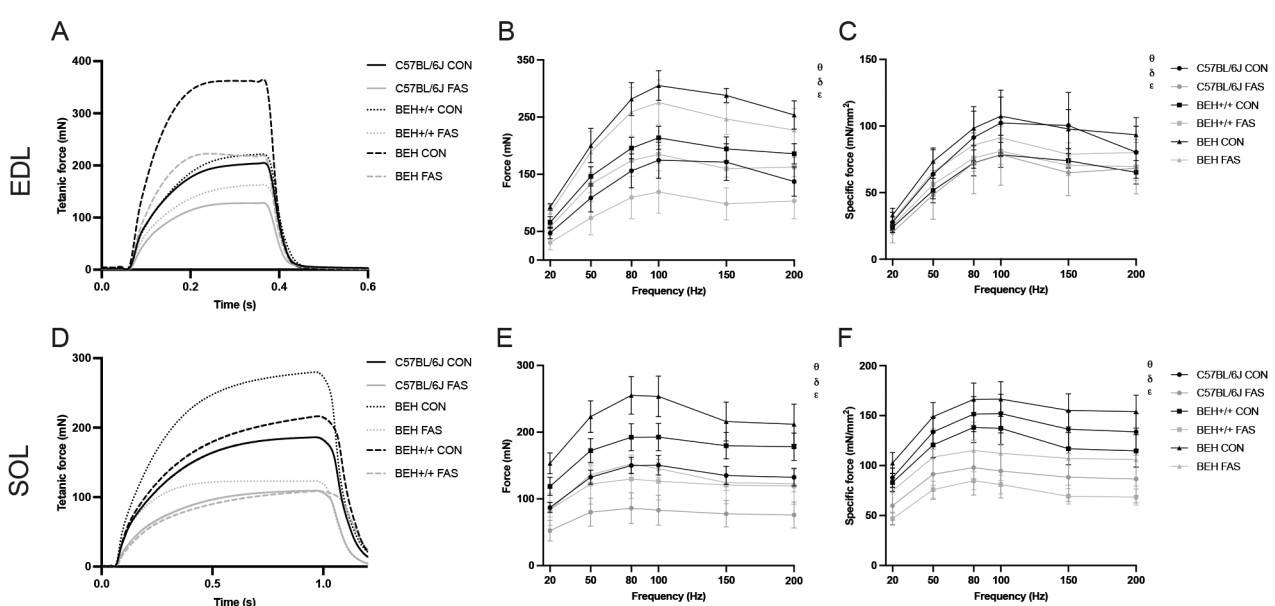

**Fig. 3. Tetanic contraction properties and force-frequency responses in EDL and SOL muscles across genotype and fasting conditions.** (A,D) Example of tetanic force-time curves at the highest stimulation frequency for representative muscles per group, (B,E) absolute force-frequency curves (mN), and (C,F) specific force-frequency curves (mN/mm²). Data are shown as mean±s.d. θ: significant condition effect, $P<0.05$; δ: significant strain effect, $P<0.05$; ε: significant muscle type effect, $P<0.05$.

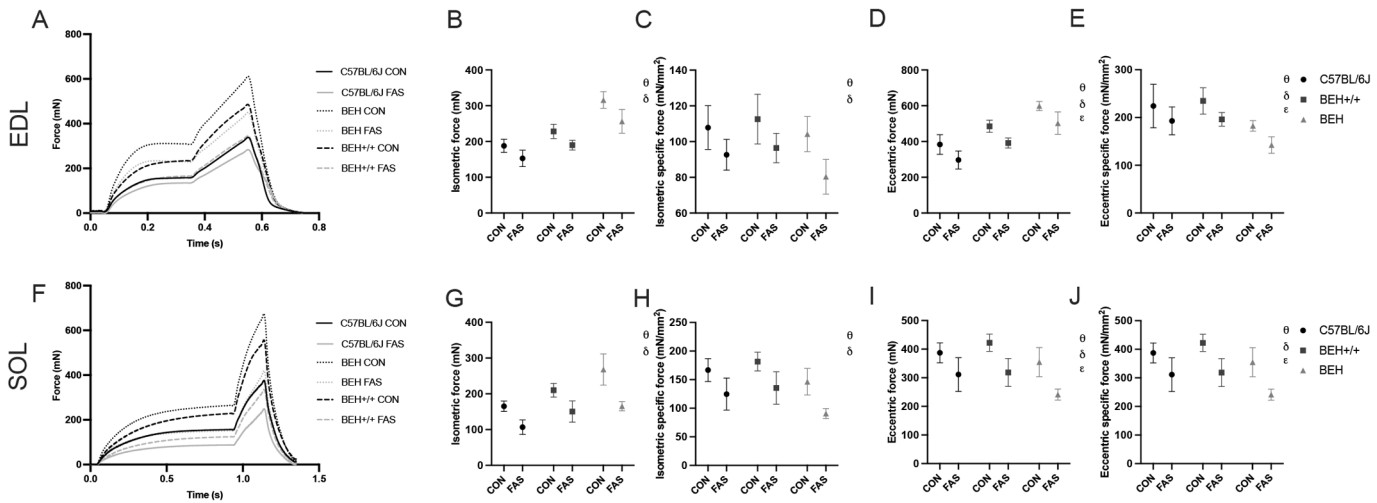

**Fig. 4. Baseline isometric and eccentric mechanical properties in EDL and SOL muscles across genotype and fasting conditions.** (A,G) Example of isometric-eccentric force-time curves, (B,H) peak absolute and (C,I) specific baseline isometric force and (D,J) peak absolute baseline eccentric force, (E,K) baseline stiffness and (F,L) tangent modulus (mean±s.d.) for EDL and SOL muscles in C57BL/6J (black dots), BEH+/+ (dark gray squares), and BEH (light gray triangles) mice, in CON and FAS groups. θ: significant condition effect, $P<0.05$; δ: significant strain effect, $P<0.05$; ε: significant muscle type effect, $P<0.05$.

any particular strain. This suggests that the catabolic response to fasting affects both slow- and fast-twitch muscles similarly, regardless of muscle phenotype differences among strains.

Previous studies have shown that the SOL muscle contains significantly greater amounts of glycogen compared to the EDL (Xu et al., 2017). However, it is likely to be more susceptible to substrate depletion during fasting than EDL as it is a postural muscle characterized by a continuous activity (Shenkman et al., 2024; Hsu et al., 2024 preprint), even in conditions of reduced physical activity as reported during fasting (Kvedaras et al., 2021). Previous studies have reported conflicting findings regarding the role of myostatin inhibition in catabolic states, with some suggesting a protective effect against muscle wasting (Lapinskas et al., 2020; Ryan et al., 2013), and others showing the opposite, such as greater muscle loss in myostatin-dysfunctional BEH mice compared to wild type during 12 weeks of 30% caloric restriction (Kvedaras et al., 2019). In our study, however, no differences were observed across strains or between muscle types in response to fasting, suggesting that the severity of complete fasting may override genotype-related variability in phenotypic adaptations (Fokin et al., 2019; Kvedaras et al., 2019).

The absence of differences between the two muscles may then be related to the overall severity of the catabolic state induced by fasting, which could override the intrinsic metabolic and functional differences between the SOL and EDL. While the SOL may benefit from the fasting-induced metabolic shift due to its greater intramyocellular lipid content (Komiya et al., 2017), this advantage could be counterbalanced by its continuous activity (Shenkman et al., 2024; Hsu et al., 2024 preprint), furthermore lack of physical activity, as reported during fasting (Kvedaras et al., 2019), might also increase loss of muscle mass. In contrast, the EDL, characterized by greater ATPase and glycolytic activity (Vinnakota et al., 2010), might experience similar atrophic effects despite its distinct metabolic profile. However, although previous studies have reported no significant differences in the rates of ATPase and glycolysis between the two muscles under resting conditions and during transient anoxia (Vinnakota et al., 2010), fasting-induced atrophy and glycogen depletion may still affect their contractility differently due to their distinct energy metabolism

profiles (Komiya et al., 2017; Nemeth et al., 1989; Hettige et al., 2020).

**Contractile function**

Our results showed that fasting significantly reduced specific force during both isometric and eccentric contractions across all strains. While the reductions were evident in both SOL (slow-twitch) and EDL (fast-twitch) muscles, the effects were slightly greater in SOL. This finding suggests that slow-twitch muscles might be more susceptible to fasting-induced impairments, potentially due to their higher levels of physical activity, reliance on oxidative metabolism and greater glycogen content (Xu et al., 2017; Komiya et al., 2017), which could make them vulnerable to glycogen depletion during fasting. These results align with prior studies showing that fasting leads to significant alterations in proteolysis-related gene expression in skeletal muscles, irrespective of baseline differences in fiber type composition (Lapinskas et al., 2020). However, despite these differences, the overall time course of decline in contractile function (1st to 100th contraction) was similar between SOL and EDL, indicating a generalized catabolic response across muscle types. This similarity suggests that fasting-induced reductions in contractile function might stem from shared mechanisms, such as reduced such as a reduced rate of ATP turnover due to limited substrate availability or alterations in excitation-contraction coupling, which could equally impact both slow- and fast-twitch fibers (Vinnakota et al., 2010; Hettige et al., 2020). Importantly, we observed that isometric force declined more steeply over the 100 cycles than eccentric force, consistent with rapid energy substrate depletion in contractile fibers, while eccentric force, reflecting the resistance of ECM, titin, and non-contractile elements, showed a slower decay (Fokin et al., 2019; Tomalka, 2023).

Fasting-induced reductions in contractile function were strain-dependent, with BEH muscles exhibiting the greatest loss in both isometric and eccentric force, followed by BEH+/+ and C57BL/6J. This strain-dependent variability highlights the influence of genetic background and muscle phenotype on the extent of functional deterioration under catabolic stress. Interestingly, while BEH

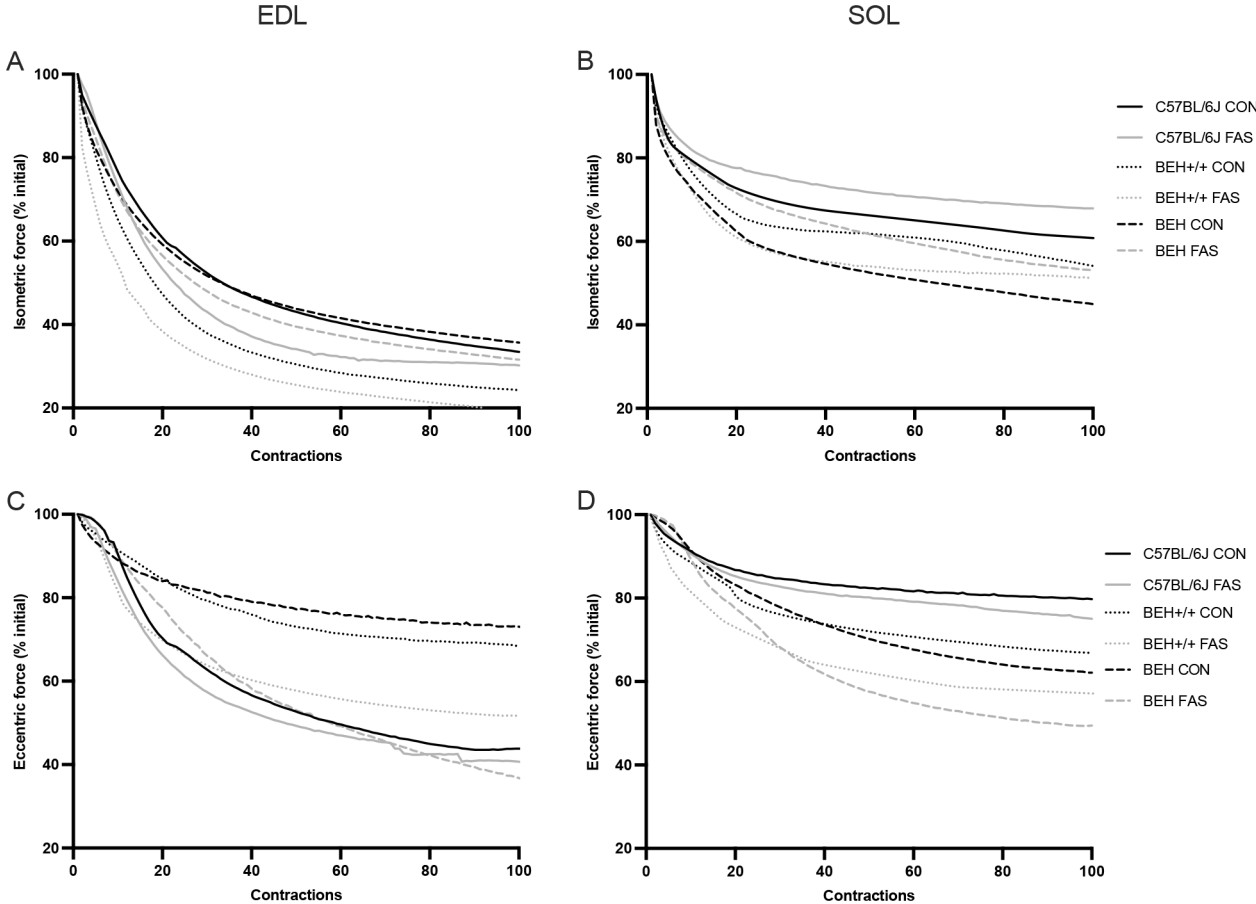

**Fig. 5. Relative isometric (A,B) and eccentric (C,D) force (mean value) normalized to baseline (1st contraction) for EDL and SOL muscles during 100 contraction cycles in C57BL/6J (continuous lines), BEH+/+ (dotted lines), and BEH (dashed lines) mice, under CON (black color) and FAS (grey color) groups.**

muscles demonstrated higher absolute values at baseline, their greater susceptibility to fasting underscores a potential trade-off between baseline functional capacity and resilience to catabolic conditions (Kvedaras et al., 2019). The larger muscle mass of BEH mice likely led to higher energetic demands during fasting, exacerbating isometric force decline, while the absence of myostatin may have contributed to reduced ECM integrity, increasing eccentric force loss (Li et al., 2008; Z Hosaka et al., 2012). BEH mice, characterized by larger muscle mass and greater strength, likely experience higher metabolic demands during fasting, which could exacerbate muscle proteolysis and functional decline (Lapinskas et al., 2020; Fokin et al., 2019; Kvedaras et al., 2019). Conversely, the relative preservation of function in C57BL/6J muscles may reflect a more balanced metabolic profile or smaller energetic demands due to their lower baseline muscle mass.

The greater reduction in eccentric-force in SOL compared to EDL suggests that slow-twitch fibers may be less equipped to handle structural and metabolic stress associated with lengthening contractions during fasting. This observation could be linked to the higher efflux of creatine kinase during mechanical stress in slow-twitch muscles, as reported by Baltusnikas et al. (2015), indicating greater susceptibility to muscle damage. Furthermore, while slow-twitch muscles are generally more resistant to fatigue due to greater reliance on oxidative metabolism compared to fast-twitch muscles, they might become less resilient under severe metabolic

stress when glycogen stores are reduced during fasting (Lapinskas et al., 2020).

However, an additional factor to consider is that BEH mouse muscles have been reported to contain a slightly higher percentage of fast-twitch fibers compared to wild-type mice (Fokin et al., 2019; Kvedaras et al., 2019). This raises the question of whether the observed effects are solely attributable to fiber-type composition or if other metabolic and structural differences between BEH and wild-type strains further modulate the response to fasting. This aspect warrants further investigation to clarify whether fiber-type composition interacts with fasting-induced muscle catabolism in myostatin-deficient models.

### Biomechanical properties

Fasting significantly reduced stiffness and tangent modulus. These reductions occurred consistently in both SOL and EDL muscles, indicating that fasting impairs muscle mechanical integrity regardless of fiber type. At baseline, SOL was characterized by greater tangent modulus, indicating stiffer and less compliant tissue behavior, and likely reflecting the intrinsic structural differences of the two muscles (James et al., 1995; Cesanelli et al., 2024).

Mouse strain-specific differences were evident in biomechanical properties, with BEH muscles generally exhibiting higher stiffness values compared to BEH+/+ and C57BL/6J. Despite having the highest baseline stiffness, BEH muscles experienced the largest reductions under fasting, particularly during eccentric contractions.

This suggests that while BEH muscles are more resilient under normal conditions, they are more susceptible to catabolic stress (i.e. prolonged fasting), potentially due to their higher baseline mechanical (i.e. force production capacity) and metabolic demands (i.e. energy requirements and turnover). Similar findings have been reported in studies examining mechanical responses to active lengthening (Satkunskiene et al., 2019), where muscles with greater baseline strength demonstrated a higher susceptibility to fatigue or damage during repeated mechanical loading.

Supporting this, myostatin-null mice exhibited reduced expression of fibroblast markers and type I collagen in tendon tissues, leading to compromised ECM structure and altered mechanical properties (e.g. strain capacity) (Mendias et al., 2008). A similar ECM dysregulation in larger/hypertrophic muscles cannot be ruled out, as myostatin signaling also regulates fibroblast activity and collagen turnover in muscle (Mendias et al., 2008; Li et al., 2008). This suggests that, in the absence of myostatin, muscles may experience structural fragility due to impaired ECM integrity, particularly under high mechanical or metabolic stress, such as during energy depletion.

The reductions in stiffness and modulus in fast- and slow-twitch muscles under fasting may stem from alterations in the ECM and sarcomere properties (Longo and Mattson, 2014; Cesanelli et al., 2024; Csapo et al., 2020). The ECM plays a critical role in maintaining tissue integrity during mechanical loading, and fasting-induced changes in proteostasis or collagen turnover could affect the stiffness and energy dissipation capacity (i.e. energy loss during loading-unloading cycles, reflecting viscoelastic properties of the tissue) of muscle-tendon units (Longo and Mattson, 2014; Kjaer, 2004). A dietary deficiency in glycine and/or proline could impair muscle connective tissue protein synthesis (Holwerda and van Loon, 2022), with evidence from chronic caloric restriction suggesting a reduction in collagen synthesis and cross-linking (Reiser et al., 1995; Sell et al., 2003). Furthermore, transcriptomic studies of EDL and SOL muscles (Hettige et al., 2020) may suggest that differences in gene expression related to structural and metabolic pathways could also contribute to the observed biomechanical changes. The consistent reduction in muscle stiffness may suggest alterations in structural elements (e.g. collagen network), together with impaired cross-bridge cycling due to limited ATP availability or proteolysis of contractile proteins (Lieber et al., 2017; Binder-Markey et al., 2021). While these interpretations remain speculative in the absence of direct structural analyses, they highlight the potential of fasting to affect both active and passive components of muscle mechanical function and potentially through distinct pathways.

Taken together, these findings highlight that while fasting induces consistent reductions in contractile function and biomechanical properties across muscle types and strains, the magnitude of these effects is in part modulated by both genetic background and muscle phenotype. Slow-twitch muscles (SOL) exhibited slightly greater impairments in force production and energy dissipation, while BEH strain muscles, despite their superior baseline function, were the most affected by fasting-induced reductions. Therefore, we confirm previous findings suggesting that food deprivation may initially induce a similar atrophic response (i.e. muscle size changes) irrespective of muscle phenotypic characteristics (Fokin et al., 2019). Moreover, we highlight that when considering contractility and mechanical function deterioration, fasting elicits a specific response that, at least in part, depends on muscle type and phenotypic traits.

These insights have practical implications for understanding how fasting-induced catabolic stress might impact physical performance and recovery in energy-restricted states, such as during prolonged fasting, caloric restriction, or illness. This is particularly relevant for activities requiring sustained or repetitive contractions, where compromised mechanical integrity could increase the risk of injury or exacerbate fatigue. Additionally, the mouse strain-dependent responses suggest that individuals with greater baseline muscle mass or strength may experience a disproportionate loss of function under fasting conditions, emphasizing the need for tailored nutritional (e.g. fasting period length personalization) and training strategies (e.g. remodulation of training intensity zones) to mitigate such effects.

### Limitations and future directions
The use of isolated muscle preparations, while enabling precise biomechanical testing, limits the ability to fully account for *in vivo* physiological and behavioral factors, such as neural input, hormonal responses, and voluntary physical activity. However, this approach also allows for the isolation of peripheral mechanisms, enabling direct assessment of intrinsic muscle properties independent of systemic influences, which is a significant methodological strength of the present work. Furthermore, because maximum shortening velocity (Vmax) was not directly measured, the applied eccentric strain rate (1.5 Lf/s) may represent different relative velocities (V/Vmax) across strains. This variability could partly influence eccentric force responses and should be considered when interpreting strain-specific differences. We did not directly assess mouse behavior or locomotion during the fasting period, and it is possible that reductions in spontaneous activity contributed to the observed impairments in muscle function. Additionally, while the 48-h fasting protocol was chosen to elicit a robust catabolic response, this duration in mice – given their high metabolic rate – may correspond to significantly longer fasting periods in humans, limiting, at least in part, its direct translational relevance. Exploring shorter fasting durations (e.g. 12–24 h) may help better model intermittent or time-restricted feeding paradigms commonly adopted in human studies. Finally, the contralateral muscle (EDL) was held in cooled Tyrode solution (4°C) without continuous $O_2$–$CO_2$ bubbling until testing, based on previous laboratory protocols and pilot experiments. While this approach preserved tissue viability, we acknowledge that continuous bubbling during this period may have further improved muscle preservation, and that this limitation could partly explain the lower stresses observed in EDL. Moreover, not all isolated muscles survived the full experimental protocol, particularly in the BEH strain, leading to smaller sample sizes in some subgroups; although the use of linear mixed models mitigated the statistical impact of such dropouts, this can be considered a potential limitation when interpreting later contraction cycles. Future research incorporating longitudinal designs, refeeding conditions, and molecular or histological endpoints will be essential to clarify the mechanisms driving these adaptations. Nonetheless, this study offers one of the first comprehensive biomechanical characterizations of skeletal muscle responses to fasting, providing a valuable experimental framework for future work.

## MATERIALS AND METHODS
### Study design
Eighteen-week-old male mice from three strains: C57BL/6J (*n*=18), BEH+/+ (*n*=20), and BEH (*n*=20), were studied. BEH mice are homozygous for the MstnCmpt-dl1Abc (Compact; Cmpt) mutation, which is associated with myostatin dysfunction. The BEH+/+ strain, carrying wild-type myostatin, was developed by crossing BEH mice with the Berlin Low (BEL) strain and repeatedly backcrossing the offspring to

BEH, using marker-assisted selection to ensure functional myostatin expression. Breeding pairs of BEH and BEH+/+ mice were generously provided by Prof. Lutz Bünger (Scotland's Rural College, UK). These strains differ markedly in muscle size – C57BL/6J exhibit normal muscle mass, BEH+/+ have moderately larger muscles, and BEH mice show markedly enlarged muscles due to myostatin deficiency – features that were expected to influence their susceptibility to fasting-induced catabolic stress.

The study was conducted in accordance with the recommendations of the Lithuanian State Food and Veterinary Service, which approved all procedures and interventions (ref. #0223 for 2012 and ref. #10 for 2014). Animals were bred and housed at the Lithuanian Sports University animal facility, maintained in standard cages (dimensions: 267×207×140 mm) at 20–21°C with 40–60% humidity and a 12/12-h reversed light/dark cycle. Mice were fed a standard chow diet (56.7% kcal carbohydrate, 29.8% kcal protein, 13.4% kcal fat; LabDiet 5001, USA) and had access to tap water *ad libitum*. At 18 weeks of age, mice from each strain were randomly assigned to either a CON group or a 48-h FAS group. This fasting duration was selected to induce substantial metabolic stress and deeper catabolic responses (Jensen et al., 2013). Mice were euthanized using gradual-fill $CO_2$ exposure. $CO_2$ was introduced into the sealed euthanasia chamber at a displacement rate of 20–30% of chamber volume per minute until loss of consciousness, followed by an additional 2–3 min to ensure complete euthanasia. All measurements were conducted at room temperature (21–23°C). Following euthanasia, the SOL and EDL muscles were carefully isolated for muscle mechanics testing.

### Experimental procedures
Mice were divided into CON and 48-h FAS groups as follows: CON: (C57BL/6J [*n*=9], BEH+/+ [*n*=10], and BEH [*n*=10]), and FAS: (C57BL/6J [*n*=9], BEH+/+ [*n*=10], and BEH [*n*=10]). The CON group had *ad libitum* access to food and water, while the FAS group had access only to water, with food withheld for 48 h. Following euthanasia, the SOL and EDL muscles were dissected, weighed with 0.1 mg precision (Kern ABS 80-4, Germany), and used for muscle force-generating capacity assessments. Muscle mechanics were examined following previously established protocols from our laboratory (Satkunskiene et al., 2019; Minderis et al., 2016; Kvedaras et al., 2017; Baltusnikas et al., 2015). After euthanasia, SOL and EDL muscles were carefully excised, with 5-0 silk sutures securely tied to their proximal and distal tendons, and placed in a 100 ml Radnoti tissue bath filled with Tyrode solution (121 mM NaCl, 5 mM KCl, 0.5 mM $MgCl_2$, 1.8 mM $CaCl_2$, 0.4 mM $NaH_2PO_4$, 0.1 mM NaEDTA, 24 mM $NaHCO_3$, 5.5 mM glucose), bubbled with a 95% $O_2$/5% $CO_2$ gas mixture and maintained at pH 7.4. The bath was kept at room temperature (21–23°C) throughout all experiments. Muscles were tested sequentially, with the EDL stored in Tyrode solution at room temperature (~21–23°C) until tested while the SOL was examined. Pilot experiments confirmed that this queuing time did not affect the muscles' force-generating capacity. For measurements, muscles were suspended vertically between two platinum plate electrodes in the bath, with the proximal tendon securely attached to the lever arm of a muscle test system (1200A-LR Muscle Test System, Aurora Scientific Inc., Canada) and the distal tendon fixed to a stable iron hook. The muscle's optimal length ($L_0$) was determined through twitch contractions, adjusting length until twitch force no longer increased. Once $L_0$ was established, the muscle was photographed against a length scale for precise length measurements (0.1 mm accuracy). During all contractile property assessments, muscles were maintained at their optimal length. After determining the $L_0$, the peak force of the muscle was assessed by stimulating it at increasing frequencies (20, 50, 80, 100, 150, and 200 Hz) using contraction durations of 0.9 s for the SOL muscle and 0.3 s for the EDL muscle. The muscle was then subjected to 100 isometric-eccentric contractions (Fig. 1) (Baltusnikas et al., 2015). For the SOL muscle, each contraction involved a total stimulation duration of 1.1 s at the frequency that generated peak force (typically 80–100 Hz), starting with an isometric phase lasting 0.9 s, followed by a 0.2-second eccentric phase during which the muscle was stretched to 30% of its optimal fiber length ($L_f$) (i.e. $1.3 \times L_f$) and then returned to its original length at the same rate (1.5 $L_f$/s) after stimulation. Similarly, for the EDL muscle, each contraction comprised a total stimulation duration of 0.5 s at the frequency that produced peak force

(typically 100 Hz), including an isometric phase of 0.3 s, followed by a 0.2-second eccentric phase where the muscle was stretched to 30% of $L_f$ (i.e. $1.3 \times L_f$) and returned at the same speed (1.5 $L_f$/s) to its original length after stimulation. Since the SOL and EDL muscles differ in their fiber attachment angles and pennation, their lengths are also different. Thus, when calculating $L_f$, we relied on the previously determined $L_f$ to $L_0$ ratios, which are 0.70 and 0.45 for the SOL and EDL muscles, respectively (Brooks and Faulkner, 1988). Furthermore, the muscle physiological cross-sectional area (PCSA) was estimated by dividing muscle wet mass by the $L_f$ and the skeletal muscle density (1.06 g/cm3) (Brooks and Faulkner, 1988). This isometric-eccentric protocol was repeated 100 times, followed by an additional isometric contraction.

### Contractile and biomechanical properties analysis
Force and displacement were analyzed using the Dynamic Muscle Analysis v5.0 software (Aurora Scientific Inc., Canada) (Cesanelli et al., 2024). Strain was subsequently calculated as a percentage of $L_0$. Stress was computed as the ratio of the loading force to the PCSA. The stiffness (ST) and tangent modulus (TM) were computed by considering the linear region of the force-displacement and stress-strain curves of the eccentric phase of the protocol (Fig. 1). All parameters were analyzed from isometric and eccentric contraction curves at 1st, 10th, 50th, 100th cycle. All tissues failing to sustain the mechanical testing protocol due to severe damages or ruptures, identifiable from real-time force–displacement traces or by visual inspection during the test, were excluded from the analyses. Additionally, force production data were analyzed for the last isometric contraction performed at the end of the 100 cycles. Peak force (PF, mN), specific force (P0, mN/mm²), rate of force development (RFD, mN/ms, calculated as the maximum slope of the force–time curve from contraction onset), specific rate of force development (sRFD, mN/ms/mm², normalized to PCSA), and PF/RFD ratio were calculated (Cesanelli et al., 2025; Barclay, 1992). Contraction time was defined as the interval from the onset of force rise to peak twitch force, and half relaxation time as the interval from peak twitch force to 50% force relaxation (Minderis et al., 2016). Fatigue index (FI%) was calculated as the percentage of force maintained from the first to the final isometric contraction. For the eccentric phases, force enhancement (stretch induced force enhancement [SIFE:*Peak eccentric force−Peak isometric force*], and residual force enhancement [RFE: *Residuale eccentric force−Peak isometric force*]) were calculated. Force–frequency data (twitch; 20–150 Hz) were analysed with tetanic force (F) modelled as a function of stimulation frequency (f) in MATLAB (R2023b, MathWorks, USA) using a three–parameter Hill–type sigmoidal function: $F(f) = \dfrac{F_{max}f^n}{(f^n + f_{50}^n)}$, where $F_{max}f^n$ is the asymptotic maximal tetanic force, $f_{50}^n$ the frequency eliciting 50% $F_{max}$ (index of activation/fusion sensitivity), and $n$ the Hill slope (curve steepness) (Brooks and Faulkner, 1988; Lieber et al., 2017; Binder-Markey et al., 2021). Initial parameter seeds (observed maximal force, median tested frequency, *n*=3) and bounded least–squares optimization (lsqcurvefit; fallback fminsearch when unavailable) were used; model quality was assessed by $R^2$ (>0.90).

### Statistical analysis
Descriptive statistics (mean±standard deviation [s.d.]) were calculated for all variables. A three-way ANOVA was performed to evaluate the main effects of mouse strain (C57BL/6J, BEH, BEH+/+), condition (CON versus FAS), and muscle type (SOL versus EDL) on muscle size parameters ($L_0$, TM, and CSA), as well as their interactions. When significant main effects were identified, Tukey's post hoc test was used for pairwise comparisons of group means. Interactions were further explored using simple effects analyses. For contractile function and biomechanical properties, data collected at multiple time points were analyzed using linear mixed models to account for repeated measures. Time (cycles), mouse strain, condition, and muscle type were modeled as fixed effects, with mouse ID included as a random effect to account for within-subject variability. Pairwise comparisons were adjusted using Bonferroni correction for multiple testing. Statistical significance was set at $P<0.05$. All analyses were conducted using IBM SPSS Statistics software for Windows® (version 28.0.0.0).

## Conclusions

Our findings demonstrate that fasting induces consistent reductions in contractile function and biomechanical properties across muscle types, with larger muscles phenotypes and slow-twitch muscles being slightly more vulnerable to these impairments. Genetic background and muscle phenotype play crucial roles in modulating the extent of functional and mechanical decline, highlighting the importance of tailored strategies to mitigate the effects of energy restriction on muscle performance.

## Competing interests

The authors declare no competing or financial interests.

## Author contributions

Conceptualization: L.C., B.Y., M.B., A.R., D.S., P.M.; Data curation: L.C., D.S.; Formal analysis: L.C., B.Y.; Investigation: L.C., P.M.; Methodology: L.C., N.E., A.R., D.S., P.M.; Project administration: M.B., P.M.; Software: B.Y., N.E.; Supervision: M.B., A.R., D.S., P.M.; Validation: D.S.; Visualization: L.C., B.Y.; Writing – original draft: L.C., B.Y., P.M.; Writing – review & editing: L.C., B.Y., M.B., N.E., A.R., D.S., P.M.

## Funding

 Deposited in PMC for immediate release.

## Data and resource availability

All relevant data and details of resources can be found within the article and its supplementary information. Additional data supporting the findings of this study are available upon request from the corresponding author.

## Peer review history

The peer review history is available online at https://journals.biologists.com/bio/lookup/doi/10.1242/bio.xxxxxx.reviewer-comments.pdf

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
