## [Peer Review File · Biology Open]

Metabolic Stress and Muscle Mechanics: Acute Response of Isolated Soleus and EDL Muscles to Prolonged Fasting in Mice with Distinct Muscle Phenotypes

Berta Ylaite, Marius Brazaitis, Nerijus Eimantas, Aivaras Ratkevicius, Danguole Satkunskiene, Petras Minderis and Leonardo Cesanelli
DOI: 10.1242/bio.062245

Editor: Lewis Halsey

Review timeline

Original submission:	7 July 2025
Editorial decision:	9 July 2025
Resubmission:	4 September 2025
Editorial decision:	12 September 2025
First revision received:	18 September 2025
Accepted:	26 September 2025

Original submission

First decision letter

MS ID#: bio.062150

MS Title: Metabolic Stress and Muscle Mechanics: Mouse-Strain-Specific Acute Mechanical Response of Isolated Soleus and EDL muscles to Prolonged Fasting

Authors: Leonardo Cesanelli; Berta Ylaite; Marius Brazaitis; Nerijus Eimantas; Aivaras Ratkevicius; Danguole Satkunskiene; Petras Minderis

Dear Dr Cesanelli,

I have now reached a decision on the above manuscript.

The reviewer reports are shown at the bottom of this email or can be accessed, together with a copy of this decision letter, by going to:

Review of your article has raised several important concerns that, together, are significant enough to prevent me from accepting it for publication. I am sorry to write with this disappointing news; however, I am sure that you appreciate that the conclusions of your research must be seen by the wider community to be fully supported by the data. My main concerns are those detailed by the first reviewer in the first part of their critique, along with the questionable fasting timescale of 48 h. On the other hand, the reviewers recognise that your study represents a lot of valuable data and if 'repackaged' could be worthy of renewed consideration.

Therefore, should you be able to carry out all the work suggested by the referees, then I would be happy to see the paper again, but as a new submission. If after considering the feedback, you instead decide to submit elsewhere, please let me know, so that we can close our file.

Reviewer 1

Thank you for the opportunity to review this manuscript. The paper is thorough and well written, however I feel that there are several considerations and limitations which are not discussed that hinder its overall relevance to human muscle physiology.

Comments are given below, ordered by subheading

Introduction

Overall, I feel that the introduction is overly long and convoluted, with several topics discussed in detail which, while relevant to the overarching subject of the impacts of metabolic stress on muscle physiology, aren't necessarily relevant to the questions being asked in this manuscript. My suggestions would be that paragraphs 2, 3 and 4 of the introduction can be condensed into one shorter paragraph, where topics such as inflammatory responses, energy supply and extra-cellular matrix remodelling can be briefly mentioned, but not discussed in too much detail. These paragraphs, as they currently are, probably fit better into the Discussion section, where your results can be put into the context of these factors.

Further to this, paragraph 5 of the introduction also needs some work. The first sentence is very long (4 lines!), and also mentions mouse muscles for the first time. This seems very out of place, when the prior paragraphs have spoken in general terms about muscle physiology and often in the context of humans. Consider condensing this paragraph to be a more general introduction to the concepts of fibre type and myostatin potentially having influences on responses to fasting, without placing it too much in the context of mouse muscles. The use of mouse models doesn't really need to be mentioned until the final paragraph of the introduction.

Methods

What is the rationale for the 48 hours fasting period? This seems extreme, given that substantial effects are seen after much shorter periods of time (as detailed by Jensen et al. <https://journals.sagepub.com/doi/full/10.1177/0023677213501659>). This is an important paper to read and cite, as it would give context into what is already known about the impacts of fasting on mice and therefore justify your experimental procedure. It would also give ideas for further discussion points surrounding the impacts of fasting on mouse behaviour (e.g. lower activity, adjusted water and non-food intake etc.), that may be seen as limitations of the study.

Further to this, did any mice not survive the 48hour fasting period? Is this why only 96 muscles out of a total of 118 were tested?

More information about the mouse strains is needed. What were the body masses of the mice, both before and after fasting? Normalising muscle body to body mass may also help, to show if the hypertrophic mice lost a greater proportion of their body mass than BL6.

Are there known differences in muscle architecture (i.e. fibre lengths), force-length or force-velocity relationships between the strains (the potential difference in fibre types is mentioned in the discussion, so I assume there are)? This will impact your experimental procedure, as you have applied the same protocol to each mouse strain. For example, I would imagine that the BEH mice have different Lf to L0 ratios than BL6 mice in both muscles. A figure showing these differences between the strains, if you know them, would be helpful.

When you say "stretched a muscle to 30% of its optimal length" I assume this means "1.3x its optimal length"?

There are several instances throughout of the f in "Lf" not being subscript.

The acronym TM refers to both "tissue mass" and "tangent modulus". Could "tissue mass" be changed to "muscle mass"?

Generally, a large portion of the Methods is given to the description of metrics, which are only presented in Supplementary Tables and which aren't mentioned in the Results or Discussion in any real detail. Can the calculation of these also be moved to the Supplementary Material?

Results

Supplementary Table 3 is not cited in the text

Discussion

Additions to the limitations paragraph discussing how the impacts of fasting on mouse behaviour (and other factors) may have affected your results would be useful. Additionally, the 48-hour fasting period may not be particularly relevant to any human contexts, given that the larger metabolic rate of mice means that this is analogous to a much longer period for humans. Fasting for different time points (e.g. 12 hours, 24 hours etc.) may be useful in future work.

Reviewer 2

My major concern is that the approach used (the Isometric-eccentric protocol) is not validated. Mainly, the authors are applying a constant strain (30%) to a tetanised muscle at a constant velocity (1.5fl/s) but they have no idea what the maximum shortening velocity (V_{max}) of the muscle is for the 6 different experimental groups. This means, that you could be stretching one muscle at a high V/V_{max} and another at a lower V/V_{max} . This is critical because the rate of force development during an eccentric contractions is velocity dependent (doi.org/10.1007/s00424-023-02794-z & doi.org/10.1113/JP285549). This is also ignoring the lack of morphometric measures to confirm that the relative muscle fibre length - to muscle length is close to the values that they cite from the literature. Fundamentally, I think specific eccentric force is not a valid metric, neither is the hysteresis measures.

I'm also unconvinced by your methods figure 1 - the eccentric ramp does not look to have the typical biphasic stretch response (doi.org/10.1007/s00424-023-02794-z & doi.org/10.1113/JP285549). I would like to see all the raw cycle 1 (i.e. each mouse) data across the six groups.

I think that the eccentric data is not valid and should not be a major part of the manuscript. There is scope however to report more data which they will have generated. (1) the isometric twitch data measured at L0 would be interesting to discuss force rise and relaxation times and the impact of fasting. (2) the force-length relationship of the six groups, it would be interesting to see if/how the passive force properties differ.

Abstract

Line 14-16, which strain do the 67% and 33% refer to? Is this a pooled average decrease across the three strains?

Line 16-17, "largest loss in contractility" what does contractility relate to specifically?

Line 18-19 "contractility of larger and slow twitch muscles" I don't think you test the impact of fasting on different size muscles, so can't make this statement.

Methods

Can you detail the CO₂ exposure used to kill the mice? How long was the exposure, the percentages used, was this a rising concentration held for a period of time?

What is Peak isometric displacement? If there is changes in length of the muscle this means that the contractions are by definition not isometric. Is this measure taken from the ergometers length control output?

What is the rational for doing any eccentric contractions ? Why have you not just done a series of cyclical shortening fatigue runs (doi.org/10.1242/jeb.243285) or simply tetanic runs (doi.org/10.1007/BF00586510). A 30% eccentric stretch is enormous and very damaging.

Page 5 - Line 22-23, you should clarify if the the resting unused muscle was left at room temperature and if it still being bubbled with Co₂/O₂ mix.

Page 5 - Line 53-55, what is the hysteresis percentage? Where is this measure taken from? Does this refer to the active/passive loops on figure 1? This needs more explaining.

Results

Page 6, Line 36-38, what does tissue failure mean? Does this mean the knot has slipped from the muscle during the eccentric ramp? I personally think tissue failure is a meaningless metric.

I can't see anywhere the report of muscle isometric properties? I think it would be very useful to have the twitch rise and relaxation times for all these groups, also what is the baseline tetanic stress? I think there should be a table of all these values. Reading from Figure 2 your EDL's generate ~100mN/mm² and the SOL 150mN/mm²? I'm concerned at the low stress in the EDL, and I think this might be because of the method of killing the mice, having the whole body exposed to rising CO₂ is likely very damaging for the muscle.

I would quite like to see the force-length relationship data for the 6 different groups, you have the data from the optimised twitches, it would give us an insight into changes in the active and passive properties of these muscles. You could present the passive force at L0 as a value (doi.org/10.1113/JP285549)

I would like to see representative figures for all mice groups of the whole cycle 1 contraction, you can overlay them like they do in this manuscripts, their figure 4 (doi.org/10.1242/jeb.197038). You can then present them in the supplementary figures.

Data presentation on the whole, I don't see why you would split up the contractions into the 1st, 10th, 50th and 100th. And having the SOL and EDL on the graphs makes them crowded and illegible. I think it would be better to present all the 100 cycles of data and have them joined with a line of best fit, and have error bars on throughout. I recommend separating graphs for the SOL and EDL. As extra encouragement to present the whole 100 cycles, it's previously been thought you can see fatigue phases during isometric tetanic contractions where fast fatigable fibres drop out and it might be you can draw attention to that in your fatigue cycles.

You could also remove any baseline differences in stress by normalising the values for each animal to cycle 1 (making it 100%) and showing the decline across the 100 cycles, you might be able to unpick more differences in the isometric contractile properties.

Supplementary Figure 1 - these data should be presented as stress not absolute force (absolute force is directly related to mass of the muscle, which you have said is different post fasting, and across mouse strains).

Conclusion

I don't think its appropriate to say that 'bigger' muscles present with greater impairments.

References for the authors to see:

doi.org/10.1242/jeb.243285 - Shelley, S. P., James, R. S., Eustace, S. J., Eyre, E., & Tallis, J. (2022). Effect of stimulation frequency on force, power and fatigue of isolated mouse extensor digitorum longus muscle. *Journal of Experimental Biology*, 225(9), jeb243285.

doi.org/10.1007/BF00586510 - Egginton, S. Effects of an anabolic hormone on striated muscle growth and performance. *Pflugers Arch.*410, 349-355 (1987).

doi.org/10.1113/JP285549 - Kissane, R. W., & Askew, G. N. (2024). Conserved mammalian muscle mechanics during eccentric contractions. *The Journal of physiology*, 602(6), 1105-1126.

doi.org/10.1007/s00424-023-02794-z - Tomalka, André. "Eccentric muscle contractions: from single muscle fibre to whole muscle mechanics." *Pflügers Archiv-European Journal of Physiology* 475.4 (2023): 421-435.

doi.org/10.1242/jeb.197038 - Tahir, U., Monroy, J. A., Rice, N. A., & Nishikawa, K. C. (2020). Effects of a titin mutation on force enhancement and force depression in mouse soleus muscles. *Journal of Experimental Biology*, 223(2), jeb197038.

Reviewer's Responses to Questions

Experimental quality

Does each figure have the proper controls?

If 'No', please indicate reasons in Comments for Author box below.

Reviewer #1:

- Yes

Reviewer #2:

- Yes

Were the data analyzed using appropriate statistical tests?

If 'No', please indicate reasons in Comments for Author box below.

Reviewer #1:

- Yes

Reviewer #2:

- Yes

Reproducibility

Were experiments performed using adequate number of biological replicates?

If 'No', please indicate reasons in Comments for Author box below.

Reviewer #1:

- Yes

Reviewer #2:

- Yes

Does the methods section provide sufficient detail to permit reproducibility?

If 'No', please indicate reasons in Comments for Author box below.

Reviewer #1:

- Yes

Reviewer #2:

- Yes

Completeness

Are the manuscript's conclusions supported by the data?

If 'No', please indicate reasons in Comments for Author box below.

Reviewer #1:

- Yes

Reviewer #2:

- Yes

Scholarship

Do the authors cite and discuss the merits of data that would argue for and against their conclusion?

If 'No', please indicate reasons in Comments for Author box below.

Reviewer #1:

- Yes

Reviewer #2:

- Yes

Does the manuscript title & abstract accurately reflect the contents of the manuscript, without hyperbole?

If 'No', please indicate reasons in Comments for Author box below.

Reviewer #1:

- Yes

Reviewer #2:

- Yes

Author response to reviewers' comments

Comments from the Reviewers:

Reviewer 1: Thank you for the opportunity to review this manuscript. The paper is thorough and well written, however I feel that there are several considerations and limitations which are not discussed that hinder its overall relevance to human muscle physiology.

Reply: We thank the reviewer for the valuable feedback provided and the time spent to revise our work. Below we provide a point by point response to the comments following the updates made to our manuscript.

Comments are given below, ordered by subheading

Introduction

Overall, I feel that the introduction is overly long and convoluted, with several topics discussed in detail which, while relevant to the overarching subject of the impacts of metabolic stress on muscle physiology, aren't necessarily relevant to the questions being asked in this manuscript. My suggestions would be that paragraphs 2, 3 and 4 of the introduction can be condensed into one shorter paragraph, where topics such as inflammatory responses, energy supply and extra-cellular matrix remodelling can be briefly mentioned, but not discussed in too much detail. These paragraphs, as they currently are, probably fit better into the Discussion section, where your results can be put into the context of these factors.

Further to this, paragraph 5 of the introduction also needs some work. The first sentence is very long (4 lines!), and also mentions mouse muscles for the first time. This seems very out of place, when the prior paragraphs have spoken in general terms about muscle physiology and often in the context of humans. Consider condensing this paragraph to be a more general introduction to the

concepts of fibre type and myostatin potentially having influences on responses to fasting, without placing it too much in the context of mouse muscles. The use of mouse models doesn't really need to be mentioned until the final paragraph of the introduction.

Reply: We appreciate this perspective and have carefully reconsidered the structure and content of the Introduction. However, given the complex nature of our model – which integrates phenotypic muscle differences, contractile and mechanical properties, and fasting-induced metabolic stress – we feel that the inclusion of key mechanistic concepts (e.g., ECM remodeling, sarcomeric integrity, energy balance) is necessary to support the study rationale. The Introduction is now down to 700 words and we tried to maintain the same logic flow and progression from general to specific information, culminating in a focused aim and hypothesis. We have nevertheless revisited the text to ensure clarity and flow, while condensing part of the sentences in the paragraphs indicated by the reviewer.

Methods

What is the rationale for the 48 hours fasting period? This seems extreme, given that substantial effects are seen after much shorter periods of time (as detailed by Jensen et al. <https://journals.sagepub.com/doi/full/10.1177/0023677213501659>). This is an important paper to read and cite, as it would give context into what is already known about the impacts of fasting on mice and therefore justify your experimental procedure. It would also give ideas for further discussion points surrounding the impacts of fasting on mouse behaviour (e.g. lower activity, adjusted water and non-food intake etc.), that may be seen as limitations of the study.

Reply: Mice assigned to the fasting group were subjected to a 48-hour food deprivation, with water available ad libitum. This duration was chosen to elicit substantial metabolic stress and muscle remodeling without compromising animal survival, based on prior studies (10.1016/j.exger.2021.111474; <https://pubmed.ncbi.nlm.nih.gov/31475942/>) and pilot testing from our lab. Although shorter fasting periods (e.g., 6-24 hours) can already alter metabolism, inflammatory tone, and muscle physiology, more prolonged fasting is known to induce deeper catabolic responses, including enhanced autophagy, increased proteolysis, and shifts in energy substrate utilization, which are relevant for the scope of this study. We have now included the suggested reference by Jensen et al. (2013) to strengthen this rationale, and we thank the reviewer for this helpful recommendation. All animals tolerated the procedure without overt signs of distress, and no mortality was observed. However, some muscles were excluded from final analyses due to mechanical failure or tissue rupture during testing, resulting in a final sample of 96 muscles out of 118 initially collected (we pointed this out later). We implemented the limitations paragraph with the points suggested by the reviewer.

Further to this, did any mice not survive the 48hour fasting period? Is this why only 96 muscles out of a total of 118 were tested?

Reply: We confirm that all mice survived the 48-hour fasting period. However, only 96 muscles were retained for analysis, as the remaining samples failed to sustain the mechanical testing protocol. In those excluded cases, severe tissue damage or partial ruptures occurred, which were clearly identifiable from real-time force-displacement traces or by visual inspection during the test. These muscles were therefore excluded to maintain data quality and consistency. We have specified this aspect in the methods.

More information about the mouse strains is needed. What were the body masses of the mice, both before and after fasting? Normalising muscle body to body mass may also help, to show if the hypertrophic mice lost a greater proportion of their body mass than BL6.

Reply: We thank the reviewer for the suggestion. We have now provided the body mass and muscle mass to body mass ratio to the first results paragraph.

Are there known differences in muscle architecture (i.e. fibre lengths), force-length or force-velocity relationships between the strains (the potential difference in fibre types is mentioned in the discussion, so I assume there are)? This will impact your experimental procedure, as you have applied the same protocol to each mouse strain. For example, I would imagine that the BEH mice

have different Lf to L0 ratios than BL6 mice in both muscles. A figure showing these differences between the strains, if you know them, would be helpful.

Reply: We thank the reviewer for the suggestion. To address the concern, we have now included full active and passive force-length (F-L) curves for both SOL and EDL across all mouse strains and experimental conditions. We would like to emphasize that our protocol was specifically designed with the primary goal of determining each muscle's optimal length (L₀), which was then used for the subsequent testing procedures, as described in the Methods section.

Regarding muscle architecture, we agree that factors such as fibre length (L_f), pennation angle, and L_f/L₀ ratios could theoretically influence muscle performance and shape the F-L curve. However, direct architectural measurements were not performed in this study. Instead, we used literature-based L_f/L₀ estimates to support physiological normalization in PCSA calculations, as previously done in related studies. Because these values were not directly measured, we did not include L_f-based analyses or architecture-specific inferences in the final analysis or interpretation.

When you say "stretched a muscle to 30% of its optimal length" I assume this means "1.3x its optimal length"? There are several instances throughout of the f in "L_f" not being subscript. The acronym TM refers to both "tissue mass" and "tangent modulus". Could "tissue mass" be changed to "muscle mass"?

Reply: we thank the reviewer for pointing out these aspects, we have now corrected and specified.

Generally, a large portion of the Methods is given to the description of metrics, which are only presented in Supplementary Tables and which aren't mentioned in the Results or Discussion in any real detail. Can the calculation of these also be moved to the Supplementary Material?

Reply: We appreciate the reviewer's suggestion and understand the concern regarding the level of detail in the Methods section. However, we chose to retain the current structure. The methodological details provided help the reader understand how specific metrics were derived, especially given the complex nature of the mechanical testing protocol. We also note that we have revised and clarified parts of the Methods section in response to reviewers suggestions, which we believe improves the overall flow and accessibility without detracting from readability. We hope the reviewer finds this compromise acceptable.

Results

Supplementary Table 3 is not cited in the text

Reply: we thank the reviewer for pointing out these aspects, we have now corrected and specified.

Discussion

Additions to the limitations paragraph discussing how the impacts of fasting on mouse behaviour (and other factors) may have affected your results would be useful. Additionally, the 48-hour fasting period may not be particularly relevant to any human contexts, given that the larger metabolic rate of mice means that this is analogous to a much longer period for humans. Fasting for different time points (e.g. 12 hours, 24 hours etc.) may be useful in future work.

Reply: We thank the reviewer for the suggestion. We have now revised the limitation paragraph, including the aspects mentioned in this comment.

Reviewer 2: My major concern is that the approach used (the Isometric-eccentric protocol) is not validated. Mainly, the authors are applying a constant strain (30%) to a tetanised muscle at a constant velocity (1.5fl/s) but they have no idea what the maximum shortening velocity (V_{max}) of the muscle is for the 6 different experimental groups. This means, that you could be stretching one muscle at a high V/V_{max} and another at a lower V/V_{max}. This is critical because the rate of force development during an eccentric contractions is velocity dependent (doi.org/10.1007/s00424-023-02794-z & doi.org/10.1113/JP285549). This is also ignoring the lack of morphometric measures to confirm that the relative muscle fibre length - to muscle length is close to the values that they cite

from the literature. Fundamentally, I think specific eccentric force is not a valid metric, neither is the hysteresis measures.

I'm also unconvinced by your methods figure 1 - the eccentric ramp does not look to have the typical biphasic stretch response (doi.org/10.1007/s00424-023-02794-z & doi.org/10.1113/JP285549). I would like to see all the raw cycle 1 (i.e. each mouse) data across the six groups.

I think that the eccentric data is not valid and should not be a major part of the manuscript. There is scope however to report more data which they will have generated. (1) the isometric twitch data measured at L0 would be interesting to discuss force rise and relaxation times and the impact of fasting. (2) the force-length relationship of the six groups, it would be interesting to see if/how the passive force properties differ.

Reply: We thank the reviewer for the constructive feedback and the time dedicated to providing suggestions to improve our work. Many of the comments indeed helped refine and clarify our approach.

Regarding the protocol used: this experimental setup has been previously validated and applied in multiple studies from our group (Satkunskiene 2019; Minderis 2016; Baltusnikas 2015), consistently producing physiologically meaningful measures of force, stiffness, and fatigue. Figure 1 illustrates the setup of our experimental protocol, while the biphasic behavior noted by the reviewer was appreciable in the real traces from isolated tissues, which we have now included as requested. Our traces may vary slightly from those in the cited works, as we tested whole muscles rather than single fibers. We also updated Figure 1 following the other revisions made to the manuscript.

We agree with the reviewer that the metric of eccentric specific force is not appropriate in this context. Normalizing eccentric force to PCSA assumes the same scaling between contractile force and passive resistance, which is not fully physiologically valid, as eccentric behavior depends largely on extracellular matrix and other non-contractile structures (e.g., titin) rather than solely on sarcomere cross-bridge capacity. We therefore removed this metric and now report only absolute eccentric force.

Similarly, we acknowledge the limitations of hysteresis measures within the present eccentric (and not passive-only) cyclic protocol, where interpretation may be confounded by fatigue and baseline drift. For this reason, hysteresis was removed from the main analyses. Instead, we retained stiffness and tangent modulus as more robust descriptors of eccentric phase mechanics (see also Satkunskiene et al., 2019).

We further refined the interpretation of the full 100-cycle data, which we decided however to report for both isometric and eccentric phases (and not isometric only). Specifically, and in line with our hypotheses and theoretical background, we interpret isometric force changes as primarily reflecting metabolic fatigue of the muscle fibers (substrate depletion and contractile dysfunction), whereas eccentric force decline may better capture fatigue or microdamage of extracellular and non-contractile structures resisting stretch after isometric contractions. This complementary analysis provides additional mechanistic insight into the distinct contributions of contractile and non-contractile tissues under repeated loading, which we further discuss in the manuscript.

Finally, while we acknowledge that the relative stretch velocity (V/V_{max}) was not directly assessed and may differ between strains, this parameter is difficult to determine and normalize in isolated muscle studies without substantially increasing experimental burden. To address this point, we added a note in the limitations, stating that variability in V/V_{max} across genotypes could influence eccentric responses.

As mentioned, raw cycle graphical traces from all groups and tests are now provided as requested.

Abstract

Line 14-16, which strain do the 67% and 33% refer to? Is this a pooled average decrease across the three strains?

Line 16-17, "largest loss in contractility" what does contractility relate to specifically?

Line 18-19 "contractility of larger and slow twitch muscles" I don't think you test the impact of fasting on different size muscles, so can't make this statement.

Reply: We have revised the abstract accordingly to clarify, for example, that the 67% and 33% values refer specifically to the fatigue index of SOL and EDL muscles, respectively, and are not pooled across strains. The term "contractility" has been replaced with "tetanic force capacity" to better reflect the parameter being measured.

Methods

Can you detail the CO₂ exposure used to kill the mice? How long was the exposure, the percentages used, was this a rising concentration held for a period of time?

Reply: We thank the reviewer for the suggestion. We have now clarified the CO₂ euthanasia procedure in the Methods section, specifying that a gradual-fill method was used at 20-30% chamber volume per minute, as per AVMA recommendations, with an additional 2-3 minutes following loss of consciousness to ensure complete euthanasia.

What is Peak isometric displacement? If there is changes in length of the muscle this means that the contractions are by definition not isometric. Is this measure taken from the ergometers length control output?

Reply: We thank the reviewer for the good point made here. In line with the concern raised, we revisited the interpretation of peak displacement during the isometric phase. We agree that any measurable change in length challenges the definition of an isometric contraction. Given this ambiguity, and uncertainty as to whether the minor compliance arose from the muscle tissue itself or the surrounding setup (e.g., suture slack or system compliance), we decided to discard this metric from biomechanical interpretation. As clarified in the revised Methods, isometric contractions are now used exclusively to assess contractile capacity (e.g., twitch and tetanic force), while mechanical and structural properties are derived from the eccentric phase of the protocol.

What is the rationale for doing any eccentric contractions? Why have you not just done a series of cyclical shortening fatigue runs (doi.org/10.1242/jeb.243285) or simply tetanic runs (doi.org/10.1007/BF00586510). A 30% eccentric stretch is enormous and very damaging.

Reply: We thank the reviewer for raising this important point. The choice of eccentric contractions over repeated stretch-shortening or tetanic fatigue runs was based on our aim to simultaneously evaluate both contractile performance and biomechanical properties of skeletal muscles under metabolic stress conditions. Eccentric loading can provide indeed a comprehensive mechanical challenge, allowing assessment of force-generating capacity, non-contractile elements properties, and structural integrity in a single experimental setup. The 30% strain amplitude was deliberately chosen—based on previous studies (10.1016/j.exger.2021.111474; <https://pubmed.ncbi.nlm.nih.gov/31475942/>) and pilot testing in our laboratory—to probe tissue fragility and the capacity to withstand mechanical stress. This is particularly relevant in fasted muscles, where metabolic remodeling may compromise structural proteins and extracellular matrix integrity. This approach enabled us to detect subtle or early signs of mechanical vulnerability without requiring separate protocols for fatigue, stretch tolerance, and passive mechanics, which would have necessitated sacrificing a larger number of animals. Thus, the applied eccentric protocol offered a high-information, ethically efficient method to characterize how different muscle types and genetic backgrounds respond to fasting-induced mechanical stress, while remaining consistent with prior literature addressing muscle damage susceptibility and energy dissipation under load. Following the reviewer's comment, we decided anyway to mention this aspect as part of the limitations of the study.

Page 5 - Line 22-23, you should clarify if the the resting unused muscle was left at room temperature and if it still being bubbled with Co₂/O₂ mix.

Reply: We thank the reviewer for the suggestion, we have now specified this aspect.

Page 5 - Line 53-55, what is the hysteresis percentage? Where is this measure taken from? Does this refer to the active/passive loops on figure 1? This needs more explaining.

Reply: We thank the reviewer for the suggestion. The hysteresis was calculated from the stretch-destretch curve (force-displacement, and later, stress-strain) of the eccentric phase of the protocol:

From previous version of the manuscript: "The hysteresis percentage (H) was calculated from the eccentric phase of the protocol (stretch-destretch phase) as: $H = RME / SME \times 100$, where RME is the returned potential deformation energy of the system (returned mechanical energy, area between the loading and unloading curve), and SME is the stored potential deformation energy of the system (stored mechanical energy, area underneath the loading curve) (Figure 1) (22)."

Following the reviewer's suggestion, and given that this parameter can be strongly affected by confounders inherent to our eccentric protocol, we decided to remove the hysteresis analysis from the results to avoid potential overinterpretation.

Results

Page 6, Line 36-38, what does tissue failure mean? Does this mean the knot has slipped from the muscle during the eccentric ramp? I personally think tissue failure is a meaningless metric.

Reply: Regarding the tissue failure: only 96 muscles were retained for analysis, as the remaining samples failed to sustain the mechanical testing protocol. In those excluded cases, severe tissue damage or partial ruptures occurred, which were clearly identifiable from real-time force-displacement traces or by visual inspection during the test. These muscles were therefore excluded to maintain data quality and consistency. We have specified this aspect more clearly in the methods.

I can't seen anywhere the report of muscle isometric properties? I think it would be very useful to have the twitch rise and relaxation times for all these groups, also what is the baseline tetanic stress? I think there should be a table of all these values. Reading from Figure 2 your EDL's generate ~100mN/mm² and the SOL 150mN/mm²? I'm concerned at the low stress in the EDL, and I think this might be because of the method of killing the mice, having the whole body exposed to rising CO₂ is likely very damaging for the muscle.

Reply: We thank the reviewer for the suggestions. In response, we have updated the Results section to report isometric contractile properties. Regarding the concern about the relatively low specific force observed in EDL muscles, we would like to clarify that the CO₂ exposure method used for euthanasia follows standard ethical procedures recommended for rodent studies and is widely adopted in skeletal muscle physiology research. Our laboratory has consistently used this method over years of experimental work without encountering systematic reductions in force or quality of isolated muscle responses. Furthermore, we ensured minimal time between euthanasia and dissection to preserve tissue viability. We also note that differences in specific force between EDL and SOL are expected due to their intrinsic contractile properties. In mice, SOL typically exhibits higher specific force despite lower absolute force, consistent with literature (e.g., <https://pubmed.ncbi.nlm.nih.gov/31475942/>). These differences likely reflect variations in the characteristics of the two different muscles, and are not attributable to experimental artifact.

I would quite like to see the force-length relationship data for the 6 different groups, you have the data from the optimised twitches, it would give us an insight into changes in the active and passive properties of these muscles. You could present the passive force at L₀ as a value (doi.org/10.1113/JP285549).

Reply: We thank the reviewer for the suggestion. We have now included full active and passive force-length (F-L) curves for both SOL and EDL across all mouse strains and experimental conditions. We would like to emphasize that our protocol was specifically designed with the primary goal of determining each muscle's optimal length (L_0), which was then used for the subsequent testing procedures, as described in the Methods section.

I would like to see representative figures for all mice groups of the whole cycle 1 contraction, you can overlay them like they do in this manuscripts, their figure 4 (doi.org/10.1242/jeb.197038). You can then present them in the supplementary figures.

Reply: We have now included representative graphs as suggested.

Data presentation on the whole, I don't see why you would split up the contractions into the 1st, 10th, 50th and 100th. And having the SOL and EDL on the graphs makes them crowded and illegible. I think it would be better to present all the 100 cycles of data and have them joined with a line of best fit, and have error bars on throughout. I recommend separating graphs for the SOL and EDL. As extra encouragement to present the whole 100 cycles, it's previously been thought you can see fatigue phases during isometric tetanic contractions where fast fatigable fibres drop out and it might be you can draw attention to that in your fatigue cycles.

You could also remove any baseline differences in stress by normalising the values for each animal to cycle 1 (making it 100%) and showing the decline across the 100 cycles, you might be able to unpick more differences in the isometric contractile properties.

Reply: Following the reviewer's comment, we updated our data presentation by: (1) showing the full 100 contraction cycles for isometric and eccentric protocols, in addition to phase-specific views for all other variables or protocols; (2) separating SOL and EDL into distinct panels for clarity; (3) for the 100 contractions, normalising values to the first contraction (100%) to remove baseline differences. We chose to present the mean values without SDs in the figure, as adding them (either as error bars or shaded areas) substantially reduced the clarity of the curves and made it difficult to distinguish group differences (see example below). The SDs are instead reported for the corresponding cycles in the supplementary tables.

Supplementary Figure 1 - these data should be presented as stress not absolute force (absolute force is directly related to mass of the muscle, which you have said is different post fasting, and across mouse strains).

Reply: We moved the supplementary figure into the main text and expressed as both absolute and specific force as suggested.

Conclusion

I don't think its appropriate to say that 'bigger' muscles present with greater impairments.

Reply: We updated the terminology in the conclusion sentences.

References for the authors to see:

doi.org/10.1242/jeb.243285 - Shelley, S. P., James, R. S., Eustace, S. J., Eyre, E., & Tallis, J. (2022). Effect of stimulation frequency on force, power and fatigue of isolated mouse extensor digitorum longus muscle. *Journal of Experimental Biology*, 225(9), jeb243285.

doi.org/10.1007/BF00586510 - Egginton, S. Effects of an anabolic hormone on striated muscle growth and performance. *Pflugers Arch.*410, 349-355 (1987).

doi.org/10.1113/JP285549 - Kissane, R. W., & Askew, G. N. (2024). Conserved mammalian muscle mechanics during eccentric contractions. *The Journal of physiology*, 602(6), 1105-1126.

doi.org/10.1007/s00424-023-02794-z - Tomalka, André. "Eccentric muscle contractions: from single muscle fibre to whole muscle mechanics." *Pflügers Archiv-European Journal of Physiology* 475.4 (2023): 421-435.

doi.org/10.1242/jeb.197038 - Tahir, U., Monroy, J. A., Rice, N. A., & Nishikawa, K. C. (2020). Effects of a titin mutation on force enhancement and force depression in mouse soleus muscles. *Journal of Experimental Biology*, 223(2), jeb197038.

Reply: We thank the reviewer for suggesting useful references.

Resubmission

First decision letter

MS ID#: bio.062245

MS Title: Metabolic Stress and Muscle Mechanics: Mouse-Strain-Specific Acute Mechanical Response of Isolated Soleus and EDL muscles to Prolonged Fasting

Authors: Leonardo Cesanelli; Berta Ylaite; Marius Brazaitis; Nerijus Eimantas; Aivaras Ratkevicius; Danguole Satkunskiene; Petras Minderis

Dear Dr Cesanelli,

I have now reached a decision on the above manuscript.

The reviewer reports are shown at the bottom of this email or can be accessed, together with a copy of this decision letter, by going to:

As you will see, the reviewers raised a number of substantial criticisms that prevent me from accepting the paper at this stage.

They suggest, however, that a revised version might prove acceptable, if you can address their concerns. If you think that you can deal satisfactorily with the criticisms on revision, I would be pleased to see a revised manuscript. We would then return it to the reviewers.

Reviewer 1

Comments for the author

Abstract - this sentence is still quite vague "contractile as well as biomechanical properties were analyzed". Specify what you measured.

Define the muscles before you abbreviate them "Fast-twitch muscles (e.g., EDL) and slow-twitch muscles (e.g., SOL) differ in their structural components" - you have introduced them now earlier in the introduction, while your original definition was in the 'study design' methods section.

Methods:

You have added the new line in the methods stating the EDL was left at room temperature while the SOL was tested. Was the EDL bubbled with O₂-CO₂ mixture? I wonder if this is why the stresses are so low in your EDL. Did you also always do the SOL first and EDL second? We typically keep our muscles bubbled in oxygenated solution at 4degrees, reducing the metabolic rate and preserving the muscles substantially longer.

You should perhaps change your abbreviations of mass to literature standards, Mb (body mass) and Mm (muscle mass). b and m subscript.

There needs to be more clarity and description of several of your metrics:

e.g. can you clarify Peak force (PF) - absolute force in mN or N's?

Rate of force development (RFD) how is this calculated specifically, is this the differential of force (dF/dt) and time? Is this from the raw force trace, or from normalised force to PCSA? Also, what is specific rate of force development? Is this normalised to Cycle 1 tetanic force, or is this normalised to the tetanic force of the 100th cycle?

CT - contraction time, is this the time spent generating force (i.e. until it completely relaxes?) or is this the force-rise time, from min force-to-peak force?

Results:

You've stated, "with BEH mice showing the highest twitch force" This needs to be more specific is this absolute twitch force (which is largely determined by mass) or is this normalised force? Because realistically, when the data are normalised (Figure 2C,H) there looks to be less clear relationship post fasting in the EDL for example (BEH absolute twitch force vs. twitch stress invert the relationship).

Minor - Figure 2A, F it would have been more useful to have all the twitches start at the same time to see the time difference in force rise and relaxation. Staggered traces are a bit harder to interpret.

You have a unit/sample frequency issue Fig 2, and elsewhere. The twitch contraction time and half relaxation time can not be 0.026 and 0.025ms (I presume this should be seconds?)

I disagree with your suggestion that the SOL generated more stress than the EDL. These various different labs categorically generated higher stresses in the mouse EDL compared to the SOL: <https://doi.org/10.1113/JP285549>, <https://doi.org/10.1242/jeb.198.2.491>, <https://doi.org/10.1152/jappphysiol.00801.2011>,

Supplementary Figure 1 - while I appreciate the authors taking on board my suggestion, it has raised more questions. Is this just a single example of one muscle for the active and passive? You have not labelled the supplementary graphs A-D so it is not immediately clear which graphs are which. I am also concerned why the force in your bottom row suddenly decline after muscle length exceeds 1 L/L₀... passive force should continue to increase as muscle length increases. So, I don't know that I believe the data you have plot to be a meaningful representation of the muscle. Also, have you normalised the passive force to 'peak passive force' or normalised it to active muscle force?

Discussion:

"Despite their superior baseline biomechanical properties" reword this.

Subheading label 'limitations'. Remove the caveat "Nonetheless, this study.." this could go into your conclusion.

Future consideration for the authors:

I think there are a lot of atypical metrics reported in this manuscript. I do unfortunately disagree very strongly with your rebuttal to my previous comment regarding your stretch-shortening protocol:

"Regarding the protocol used: this experimental setup has been previously validated and applied in multiple studies from our group (Satkunskiene 2019; Minderis 2016; Baltusnikas 2015), consistently producing physiologically meaningful measures of force, stiffness, and fatigue. "

It is fundamentally not a validated approach (nor physiologically meaningful) if you are subjecting muscles to different relative rates of eccentric loading (because you have not quantified their maximum rate of shortening) and because you do not know the relative fibre length of the muscles the strains you are applying are unknown and could be very different across these groups. There is no paper to my knowledge that has shown the rate of fatigue is comparable when subjecting muscles to different eccentric loading rates (or strain amplitudes).

We know that if you use cyclical loading at different frequencies the fatigue response of skeletal muscle varies substantially (<https://doi.org/10.1242/jeb.200.22.2907>). This is a potential problem you have here if the muscles have a different Vmax and subsequently different optimal cyclical frequencies to generate peak power. Your figure 4A shows nicely the lengthening profiles of the EDL muscle, and if you look at the two-phase response across your six groups, there looks to be a difference in the time spent during the phase-1 (cross-bridge dependent) component before transitioning to the phase-2 (non-parallel elastic components, like titin). I would like to specifically draw your attention to the BL6 CON vs. BEH CON. This difference in time to transition is in part determined by the initial overlap of actin and myosin, which means there it is highly unlikely that these two muscles are sitting on the same force-length position (I expect partly because their muscle architecture, fibre lengths, are different).

For future experiments I would encourage you to change your protocols. The force-frequency data does not provide a vast insight into the muscle function, and I believe it takes up time which would be better spent doing an isotonic afterload force-velocity relationship. This requires a minimum of 9 contractions (1x control tetanus, 3 shortening velocities, 1x control tetanus, 3 shortening velocities, and a final control tetanus) which will provide you with a complete shortening force-velocity relationship giving you substantially more information about the ability of the muscle to shorten and generate power. It then would provide a foundation for you to do more meaningful eccentric contractile protocols, with lengthening velocities directly related to shortening velocities. I would encourage measuring muscle fibre lengths of these muscles also (more so in the GM mice), using the nitric acid digestion method you would be able to very quickly quantify the relative muscle fibre length to whole muscle length, required to correct shortening/lengthening velocities, PCSA estimates and specific force.

Finally, Figure 3D - Your tetani look to be struggling to plateau, which suggests that they are at too long lengths on the force-length curve.

Reviewer 2

Comments for the author

The authors conducted an experimental study whereby three different mouse strains, which varied in relation to their muscle masses, were subjected to a 48 hour fast, with outcomes of interest including muscle size along with various analyses related to contractile function and biomechanical properties. The authors reported that fasting induced various impairments, including reductions in muscle size, twitch force and force-frequency, isometric and eccentric contractile function and stiffness and tangent modulus. Response to fasting appeared to be influenced by both muscle type and mouse strain, with larger impairments to contractile function observed in both the soleus (slow-twitch) and BEH (myostatin deficient, hypertrophic) mice.

Overall I found this to be an interesting and well-written study and have only minor suggestions for the authors consideration.

1. If feasible, it might be interesting to make the title more informative by including a brief descriptor of the mouse strain characteristics in the title, e.g., "Metabolic Stress and Muscle Mechanics: Acute Mechanical Response of Isolated Soleus and EDL Muscle to Prolonged Fasting in Mice Strains with Varying Body Compositions" (or something along these lines).
2. Page 2, Line 10-11: Here and throughout the manuscript it would be useful to be more consistent in the way that the mouse strains are described, considering that "larger muscles" and "hypertrophic muscles" might be interpreted as being the same thing.
3. Introduction: It would be useful to provide some more contextualization around the length and type of fast you are referring to in the opening paragraph. The term "prolonged fasting" is somewhat ambiguous, and more detail on the specific durations and forms of fasting would be useful. I believe this is important because the length of fast is likely to markedly influence its biological or performance effects. I also suggest differentiating caloric restriction from fasting (namely a complete absence of food), as these are distinct interventions with distinct biological effects, and as such should not be described as interchangeable, as implied on Page 2, Line 10 - 11.
4. Further to the above point, it would be helpful to provide some more context about the severity and duration of the 48 hour fast investigated. I recognize that a 48 hour fast will provide a far greater stress in mice compared to humans, given their faster metabolic rate and life pace. This is briefly alluded to in the methods on Page 4, lines 32 - 33, and the reference to the Jensen paper was useful, however I believe it would be useful to be more explicit on this point and to consider its likely practical application to a human context, particularly in relation to the interventions described in the introduction.
5. Page 3, Paragraph starting line 40. I suggest some more care and attention to detail is required in this paragraph. Metabolic stress and prolonged inactivity are very different stressors, and results from one type of manipulation cannot be applied to the others. For example, on lines 42-43 the authors state that costameric proteins may be affected by reduced energy availability, impairing the mechanical integrity of the muscle, however neither of the two citations investigate energy availability and so it is not clear how this conclusion was reached.
6. Page 3, Line 51 - 55: I was confused by this sentence and rephrasing to enhance clarity may be useful. What did you mean by "under physiological conditions" here? If referring to an unfasted state or with adequate energy availability I suggest just saying that.
7. Page 3, Line 58 - 59: I am not entirely clear on the argument being made here. Would higher glycogen content, combined with the enhanced fat oxidation capacity of slow-twitch muscle not make them less, rather than more, vulnerable to metabolic stress during fasting? Furthermore, while some mouse studies may have reported higher glycogen in the soleus compared to the EDL, the general pattern in humans is the opposite, with fast-twitch fibres typically storing more glycogen than slow-twitch. Finally, although the SOL is more continuously active, the low intensity nature of this activity means it is likely to be mainly fueled by lipid oxidation. Considering these points, I am not entirely convinced by the arguments made regarding why the soleus may be more susceptible to metabolic stress than the EDL.
8. Methods - study design section: I suggest that this section be shortened and focused entirely on giving a broad overview of the main study design. I also suggest including a specific sub-section to further elaborate the development and characteristics of the different mouse strains investigated, along with a comment on the likely implications of these characteristics on the response to fasting.
9. Page 6, Line 12-13: Some more detail on the missing muscle samples would be useful. I assume that at least 116 isolated muscles should theoretically have been available (58 animals in the experiment, with 2 muscles extracted from each one), so it is not clear why only 96 were available for testing. While I recognize that some extent of tissue loss or failure is the reality in studies like this, the extent is concerning, considering how small some of the final sample sizes for

certain subgroups and outcomes were, e.g., it appears that only a single EDL muscle was tested for the 100th contraction of the fasted BEH group. The varying sample sizes described in Table 2 may impact statistical power and capacity to interpret these results, and as such I suggest that this could be highlighted as a limitation of the study.

10. Discussion: Overall, some very interesting points were made throughout the discussion and I appreciate the authors clear attempts to be balanced, and when speculating on potential mechanisms, that they made this clear.

11. Page 7, Line 55-57: Unless I missed it, this "trend" was not mentioned in the results and looking at the results in Table 2, I am not clear on where this came from. If the authors consider this to be an important point to make, they should elaborate on why that is, otherwise I suggest just removing this sentence.

12. Page 8, Line 6-7: In what way do these results indicate that myostatin inhibition could counteract muscle wasting in a catabolic state, considering that there was no difference for these outcomes between the strains?

13. Page 8, Line 39 - 40: Considering that cells store only limited ATP regardless of feeding state, the key issue during fasting may relate to a reduced rate of ATP turnover due to limited substrate availability to sustain contractile function, rather than to low ATP availability.

14. Page 10, Line 24 - 25: I appreciate the authors consideration of the limitations of their approach, but I also think that it is worth commenting on the benefits of examining isolated muscle preparations. I agree that this does not allow for consideration of full in vivo physiological and behavioral factors, but it also allows for isolation of peripheral mechanisms which I believe to be very valuable. Every methodological decision has a cost, and while I agree with the authors that it is useful to consider these costs, I also believe it useful to also comment on the strengths of the approach.

Reviewer's Responses to Questions

Experimental quality

Does each figure have the proper controls?

If 'No', please indicate reasons in Comments for Author box below.

Reviewer #1:

- Yes

Reviewer #2:

- Yes

Were the data analyzed using appropriate statistical tests?

If 'No', please indicate reasons in Comments for Author box below.

Reviewer #1:

- Yes

Reviewer #2:

- Yes

Reproducibility

Were experiments performed using adequate number of biological replicates?

If 'No', please indicate reasons in Comments for Author box below.

Reviewer #1:

- Yes

Reviewer #2:

- Yes
-

Does the methods section provide sufficient detail to permit reproducibility?

If 'No', please indicate reasons in Comments for Author box below.

Reviewer #1:

- Yes

Reviewer #2:

- Yes
-

Completeness

Are the manuscript's conclusions supported by the data?

If 'No', please indicate reasons in Comments for Author box below.

Reviewer #1:

- Yes

Reviewer #2:

- Yes
-

Scholarship

Do the authors cite and discuss the merits of data that would argue for and against their conclusion?

If 'No', please indicate reasons in Comments for Author box below.

Reviewer #1:

- Yes

Reviewer #2:

- Yes
-

Does the manuscript title & abstract accurately reflect the contents of the manuscript, without hyperbole?

If 'No', please indicate reasons in Comments for Author box below.

Reviewer #1:

- Yes

Reviewer #2:

- Yes

First revision

Author response to reviewers' comments

Reviewer 1:

We thank the reviewer once again for the time invested in reviewing our manuscript and for the valuable suggestions provided, not only related to the manuscript itself but also for the considerations directed to our future works. Overall we appreciated this constructive feedback.

Abstract - this sentence is still quite vague "contractile as well as biomechanical properties were analyzed". Specify what you measured.

Reply: We have revised the sentence in the Abstract to specify the measured parameters.

Isolated SOL and EDL were subjected to 100 isometric-eccentric contraction cycles, and peak and specific force, rate of force development, fatigue, stiffness, and tangent modulus were assessed.

Define the muscles before you abbreviate them "Fast-twitch muscles (e.g., EDL) and slow-twitch muscles (e.g., SOL) differ in their structural components" - you have introduced them now earlier in the introduction, while your original definition was in the 'study design' methods section.

Reply: e have revised the sentence in the Introduction to define the muscles before abbreviating, which now reads: Extensor digitorum longus (EDL; fast-twitch) and soleus (SOL; slow-twitch) muscles differ in their structural components (20), contractile behavior (21), and mechanical properties (22,23).

Methods:

You have added the new line in the methods stating the EDL was left at room temperature while the SOL was tested. Was the EDL bubbled with O₂-CO₂ mixture? I wonder if this is why the stresses are so low in your EDL. Did you also always do the SOL first and EDL second? We typically keep our muscles bubbled in oxygenated solution at 4degrees, reducing the metabolic rate and preserving the muscles substantially longer.

Reply: We thank the reviewer for this important point. In our setup, the SOL was always tested first, while the contralateral EDL was maintained in Tyrode solution at 4 °C until testing. The holding solution was not bubbled with O₂-CO₂ during this interval. This procedure was based on previous work from our laboratory and pilot experiments, where no significant reduction in muscle force was observed under these conditions. Nevertheless, we agree that continuous bubbling during the holding phase could further improve muscle preservation and acknowledge that this may have contributed to the relatively low stresses in EDL compared with SOL. We have now added this aspect to the limitations section, and we will keep this suggestion in mind for future experiments.

... Finally, the contralateral muscle (EDL) was held in cooled Tyrode solution (4 °C) without continuous O₂-CO₂ bubbling until testing, based on previous laboratory protocols and pilot experiments. While this approach preserved tissue viability, we acknowledge that continuous bubbling during this period may have further improved muscle preservation and that this limitation could partly explain the lower stresses observed in EDL...

You should perhaps change your abbreviations of mass to literature standards, Mb (body mass) and Mm (muscle mass). b and m subscript.

Reply: we have now updated the abbreviations of mass as suggested in the text and table 1.

There needs to be more clarity and description of several of your metrics: e.g. can you clarify Peak force (PF) - absolute force in mN or N's? Rate of force development (RFD) how is this calculated specifically, is this the differential of force (dF/dt) and time? Is this from the raw force trace, or from normalised force to PCSA? Also, what is specific rate of force development? Is this normalised to Cycle 1 tetanic force, or is this normalised to the tetanic force of the 100th cycle?

Reply: We thank the reviewer for this comment. All measurement units were already reported in the figures and tables, and we have now added them in the Methods section for clarity. Peak force is expressed as absolute force (mN). RFD was calculated as the maximum slope of the raw force-time curve ($\Delta F/\Delta t$) from contraction onset during each isometric phase of the eccentric protocol. Specific RFD was normalized to PCSA. The revised Methods section now states:

“Peak force (PF, mN), specific force (P0, mN/mm²), rate of force development (RFD, mN/ms, calculated as the maximum slope of the force-time curve from contraction onset), specific rate of force development (sRFD, mN/ms/mm², normalized to PCSA), and PF/RFD ratio were calculated (30,31).”

CT - contraction time, is this the time spent generating force (i.e. until it completely relaxes?) or is this the force-rise time, from min force-to-peak force?

Reply: We thank the reviewer for pointing this out. Contraction time (CT) was defined as the interval from the onset of force rise to peak twitch force, while half relaxation time (HRT) was defined as the interval from peak twitch force to 50% force relaxation, in line with standard definitions. We specified this in the methods now. We also acknowledge that there was a labeling mistake in the x-axis of the figures where time was reported, and we have now corrected this. We thank the reviewer for identifying this oversight (pointed in later comment).

“Contraction time was defined as the interval from the onset of force rise to peak twitch force, and half relaxation time as the interval from peak twitch force to 50% force relaxation (32).”

Results:

You've stated, "with BEH mice showing the highest twitch force" This needs to be more specific is this absolute twitch force (which is largely determined by mass) or is this normalised force? Because realistically, when the data are normalised (Figure 2C,H) there looks to be less clear relationship post fasting in the EDL for example (BEH absolute twitch force vs. twitch stress invert the relationship).

Reply: We thank the reviewer for noting this inconsistency. Upon re-checking our raw data, we identified a reporting error in the graph for BEH EDL, which caused the mismatch with the Results text. This has now been corrected, and the revised figure and Results section accurately reflect the data. We apologize for this oversight and thank the reviewer for helping us improve the clarity and accuracy of the manuscript. Figure 2 has been updated and the text made more clear.

Minor - Figure 2A, F it would have been more useful to have all the twitches start at the same time to see the time difference in force rise and relaxation. Staggered traces are a bit harder to interpret.

Reply: We thank the reviewer for pointing this out. We updated the Figure 2 panels A and F accordingly.

You have a unit/sample frequency issue Fig 2, and elsewhere. The twitch contraction time and half relaxation time can not be 0.026 and 0.025ms (I presume this should be seconds?)

Reply: We thank the reviewer for pointing this out. We corrected the time unit of measure in figure 1 and figure 2 accordingly.

I disagree with your suggestion that the SOL generated more stress than the EDL. These various different labs categorically generated higher stresses in the mouse EDL compared to the SOL: <https://doi.org/10.1113/JP285549>, <https://doi.org/10.1242/jeb.198.2.491>, <https://doi.org/10.1152/jappphysiol.00801.2011>,

Reply: We thank the reviewer for pointing this out and for providing references that help us to interpret and compare our findings. We acknowledge that this discrepancy may, at least in part, be related to differences in experimental conditions compared to the studies cited, including—as noted above—the fact that the EDL was held while the SOL was tested first. We now recognize this as a limitation and have added it to the revised “Limitations and future directions” section. This clarification helps to avoid overgeneralization and ensures that our finding is presented as specific to the conditions of our protocol.

Supplementary Figure 1 - while I appreciate the authors taking on board my suggestion, it has raised more questions. Is this just a single example of one muscle for the active and passive?

You have not labelled the supplementary graphs A-D so it is not immediately clear which graphs are which. I am also concerned why the force in your bottom row suddenly decline after muscle length exceeds 1 L/L₀... passive force should continue to increase as muscle length increases. So, I don't know that I believe the data you have plot to be a meaningful representation of the muscle. Also, have you normalised the passive force to 'peak passive force' or normalised it to active muscle force?

Reply: We thank the reviewer for these helpful observations. Upon re-checking, we identified a mistake in the reporting of the passive force-length data, which has now been corrected. The figure has been relabelled (panels A-D) and clarified in the legend. The traces shown represent data from a single muscle as an indicative example for each group. Passive force was normalized to peak passive force (panels C-D), while active force was normalized to peak active force (panels A-B). As the reviewer notes (and explained in our previous reply letter), our protocol was not designed to generate a full force-length relationship, but rather to determine optimal muscle length (L₀) for subsequent testing. For this reason, the length increments and overall design were not optimized for detailed force-length analysis. We better clarified all these aspect in the figure's caption.

Discussion:

"Despite their superior baseline biomechanical properties" reword this.

Reply: We thank the reviewer for the suggestion, we have reworded the sentence as:

Despite having the highest baseline stiffness, BEH muscles experienced the largest reductions under fasting, particularly during eccentric contractions.

Subheading label 'limitations'. Remove the caveat "Nonetheless, this study.." this could go into your conclusion.

We thank the reviewer for this suggestion. We have added the subheading “Limitations and Future Directions” for clarity. We decided to retain the sentence “Nonetheless, this study offers one of the first comprehensive biomechanical characterizations of skeletal muscle responses to fasting” within this section, as it emphasizes the value of our approach in the context of the discussed limitations. Retaining this statement highlights the contribution of our work and its relevance for future research, while still addressing methodological caveats. This approach is also in line with Reviewer 2's suggestion to acknowledge the strengths of the methodology alongside its limitations.

Future consideration for the authors:

I think there are a lot of atypical metrics reported in this manuscript. I do unfortunately disagree very strongly with your rebuttal to my previous comment regarding your stretch-shortening protocol:

"Regarding the protocol used: this experimental setup has been previously validated and applied in multiple studies from our group (Satkunskiene 2019; Minderis 2016; Baltusnikas 2015), consistently producing physiologically meaningful measures of force, stiffness, and fatigue. "

It is fundamentally not a validated approach (nor physiologically meaningful) if you are subjecting muscles to different relative rates of eccentric loading (because you have not quantified their maximum rate of shortening) and because you do not know the relative fibre length of the muscles the strains you are applying are unknown and could be very different across these groups. There is no paper to my knowledge that has shown the rate of fatigue is comparable when subjecting muscles to different eccentric loading rates (or strain amplitudes).

We know that if you use cyclical loading at different frequencies the fatigue response of skeletal muscle varies substantially (<https://doi.org/10.1242/jeb.200.22.2907>). This is a potential problem you have here if the muscles have a different Vmax and subsequently different optimal cyclical frequencies to generate peak power. Your figure 4A shows nicely the lengthening profiles of the EDL muscle, and if you look at the two-phase response across your six groups, there looks to be a difference in the time spent during the phase-1 (cross-bridge dependent) component before transitioning to the phase-2 (non-parallel elastic components, like titin). I would like to specifically draw your attention to the BL6 CON vs. BEH CON. This difference in time to transition is in part determined by the initial overlap of actin and myosin, which means there it is highly unlikely that these two muscles are sitting on the same force-length position (I expect partly because their muscle architecture, fibre lengths, are different).

For future experiments I would encourage you to change your protocols. The force-frequency data does not provide a vast insight into the muscle function, and I believe it takes up time which would be better spent doing an isotonic afterload force-velocity relationship. This requires a minimum of 9 contractions (1x control tetanus, 3 shortening velocities, 1x control tetanus, 3 shortening velocities, and a final control tetanus) which will provide you with a complete shortening force-velocity relationship giving you substantially more information about the ability of the muscle to shorten and generate power. It then would provide a foundation for you to do more meaningful eccentric contractile protocols, with lengthening velocities directly related to shortening velocities. I would encourage measuring muscle fibre lengths of these muscles also (more so in the GM mice), using the nitric acid digestion method you would be able to very quickly quantify the relative muscle fibre length to whole muscle length, required to correct shortening/lengthening velocities, PCSA estimates and specific force.

Finally, Figure 3D - Your tetani look to be struggling to plateau, which suggests that they are at too long lengths on the force-length curve.

Reply: We appreciate the reviewer's constructive suggestions regarding future experiments, that will be surely considered and which could help guide our ongoing and follow-up studies. The feedback from this and previous revisions has already been valuable in improving the clarity and scope of our work. We acknowledge the concerns raised regarding the stretch-shortening protocol and the limitations inherent to the current approach, including differences in relative eccentric loading rates, potential variations in fibre length, and interpretation of fatigue responses. We will carefully consider the suggested modifications, including the use of isotonic force-velocity protocols and direct measurement of fibre lengths, in future experiments. Regarding Figure 3D, we note the reviewer's observation about the plateauing of tetani, and we will take this into account when designing subsequent experiments to optimize muscle length settings and contraction parameters.

Overall, we thank the reviewer again for providing guidance that will strengthen future work and help contextualize the results presented in the current manuscript.

Reviewer 2: The authors conducted an experimental study whereby three different mouse strains, which varied in relation to their muscle masses, were subjected to a 48 hour fast, with outcomes of interest including muscle size along with various analyses related to contractile function and biomechanical properties. The authors reported that fasting induced various impairments, including reductions in muscle size, twitch force and force-frequency, isometric and eccentric contractile function and stiffness and tangent modulus. Response to fasting appeared to be influenced by both muscle type and mouse strain, with larger impairments to contractile function observed in both the soleus (slow-twitch) and BEH (myostatin deficient, hypertrophic) mice.

Overall I found this to be an interesting and well-written study and have only minor suggestions for the authors consideration.

Reply: We thank the reviewer for the time dedicated to reviewing our manuscript, for the constructive suggestions, and for the positive feedback. Below we address each comment point by point.

1. If feasible, it might be interesting to make the title more informative by including a brief descriptor of the mouse strain characteristics in the title, e.g., "Metabolic Stress and Muscle Mechanics: Acute Mechanical Response of Isolated Soleus and EDL Muscle to Prolonged Fasting in Mice Strains with Varying Body Compositions" (or something along these lines).

Reply: We thank the reviewer for the suggestion. The revised title now reads:

“Metabolic Stress and Muscle Mechanics: Acute Response of Isolated Soleus and EDL Muscles to Prolonged Fasting in Mice with Distinct Muscle Phenotypes“

2. Page 2, Line 10-11: Here and throughout the manuscript it would be useful to be more consistent in the way that the mouse strains are described, considering that "larger muscles" and "hypertrophic muscles" might be interpreted as being the same thing.

Reply: We thank the reviewer for the suggestion. We have revised the terminology to ensure consistency throughout the manuscript and avoided the term hypertrophic, as it may not be the most accurate descriptor for myostatin-deficient muscles. Specifically, we now describe the three strains as:

C57BL/6J - normal-sized muscles,

BEH+/+ - larger muscles,

BEH - myostatin-deficient mice with markedly larger muscles.

3. Introduction: It would be useful to provide some more contextualization around the length and type of fast you are referring to in the opening paragraph. The term "prolonged fasting" is somewhat ambiguous, and more detail on the specific durations and forms of fasting would be useful. I believe this is important because the length of fast is likely to markedly influence its biological or performance effects. I also suggest differentiating caloric restriction from fasting (namely a complete absence of food), as these are distinct interventions with distinct biological effects, and as such should not be described as interchangeable, as implied on Page 2, Line 10 - 11.

Reply: We thank the reviewer for the suggestion. We updated the first paragraph of the introduction as follows:

...“In clinical settings, chronic caloric restriction (i.e., reduced but sustained food intake), but also transient or complete fasting (i.e., total absence of food intake), are often employed to manage different clinical conditions (4). For instance, prolonged fasting (6 days) has recently been the focus of human trials investigating its effects on metabolism, insulin secretion, and oxidative stress (5,6).”...

4. Further to the above point, it would be helpful to provide some more context about the severity and duration of the 48 hour fast investigated. I recognize that a 48 hour fast will provide a far greater stress in mice compared to humans, given their faster metabolic rate and life pace. This is briefly alluded to in the methods on Page 4, lines 32 - 33, and the reference to the Jensen paper was useful, however I believe it would be useful to be more explicit on this point and to consider its likely practical application to a human context, particularly in relation to the interventions described in the introduction.

Reply: We thank the reviewer for the suggestion. Our aim was to study a condition that, while relatively more severe in mice (see Jensen et al. 2010 but also Rowland, Comp Med 2007; 57: 149-160) allows us to investigate the impact of prolonged fasting on muscle contractility and mechanical properties—a topic not yet fully explored in humans. As noted by the reviewer and in our Methods and Discussion sections, a 48-hour fast represents a greater physiological stress in mice due to their faster metabolism and shorter lifespan. Nevertheless, prolonged fasting in humans (including “challenging” protocols, sometimes extending up to a week or more of food deprivation) is already applied in research, clinical contexts, and wellness practices, occasionally in combination with physical exercise. Evidence on its effects on skeletal muscle, however, remains limited. We have attempted to highlight and contextualize this aspect throughout the manuscript, while other translational aspects were beyond the scope of the present work.

5. Page 3, Paragraph starting line 40. I suggest some more care and attention to detail is required in this paragraph. Metabolic stress and prolonged inactivity are very different stressors, and results from one type of manipulation cannot be applied to the others. For example, on lines 42-43 the authors state that costameric proteins may be affected by reduced energy availability, impairing the mechanical integrity of the muscle, however neither of the two citations investigate energy availability and so it is not clear how this conclusion was reached.

Reply: We thank the reviewer for this comment. We agree that metabolic stress and prolonged inactivity are distinct stressors. In our manuscript, the reference to inactivity reflects the effect of fasting, which typically reduces voluntary activity and movement. Regarding costameric proteins, their function in force transmission and sarcomere anchoring requires both ATP and cross-bridge cycling. While the cited studies did not directly examine energy availability, they support a logical framework in which inadequate energy supply may affect overall muscle function, not only via direct ATP dependence but also indirectly through impaired assembly, maintenance, or breakdown of these complexes. Similarly, sarcomeric proteins such as titin may be influenced by post-translational modifications, which can be modulated by fasting-induced oxidative or metabolic stress. We have clarified this reasoning in the revised paragraph to better reflect these mechanistic links while remaining consistent with the cited literature.

„Although a few studies have explored this aspect, fasting may also affect also the biomechanical properties of muscles, by altering their structural and molecular components. Metabolic stress and prolonged physical inactivity— a potential consequence of stressful conditions such as low energy availability—may trigger extracellular matrix (ECM) remodeling (19-21). At the cellular level, costameric proteins, which anchor sarcomeres to the sarcolemma and facilitate lateral force transmission, rely on adequate energy supply for proper assembly and function; insufficient energy could therefore indirectly impair their function and the mechanical integrity of the muscle (22,23). Similarly, sarcomeric proteins such as titin, that modulates stiffness and elastic recoil, could be compromised by post-translational modifications such as phosphorylation-dephosphorylation, which may be disrupted due to fasting-related ATP depletion and inflammatory stress (24,25).“

6. Page 3, Line 51 - 55: I was confused by this sentence and rephrasing to enhance clarity may be useful. What did you mean by "under physiological conditions" here? If referring to an unfasted state or with adequate energy availability I suggest just saying that.

Reply: We thank the reviewer for the suggestion. We clarified this point as indicated now in the introduction.

7. Page 3, Line 58 - 59: I am not entirely clear on the argument being made here. Would higher glycogen content, combined with the enhanced fat oxidation capacity of slow-twitch muscle not make them less, rather than more, vulnerable to metabolic stress during fasting? Furthermore,

while some mouse studies may have reported higher glycogen in the soleus compared to the EDL, the general pattern in humans is the opposite, with fast-twitch fibres typically storing more glycogen than slow-twitch. Finally, although the SOL is more continuously active, the low intensity nature of this activity means it is likely to be mainly fueled by lipid oxidation. Considering these points, I am not entirely convinced by the arguments made regarding why the soleus may be more susceptible to metabolic stress than the EDL.

Reply: We thank the reviewer for this comment and perspective given. We actually agreed with the points raised and reconsidered our assumptions in that passage of the introduction. In our revised paragraph, we clarified that the susceptibility of the soleus (SOL) to fasting-induced deficits is likely multi-factorial, arising not solely from glycogen depletion and activity level/characteristics, but also from structural features, and higher metabolic demands in larger muscles. We believe this revised text better reflects the mechanistic rationale without overgeneralizing from glycogen content alone.

“...Fasting may further exacerbate these phenomena through accelerated proteolysis, glycogen depletion, and altered extracellular matrix remodeling. While the metabolic response to fasting may differ between fiber types—with slow-twitch muscles relying more on lipid oxidation and fast-twitch muscles being more sensitive to glycogen depletion—SOL muscles may still experience vulnerability due to structural features, pre-existing myostatin dysfunction, and higher metabolic demands associated with larger muscle size. Collagen degradation and ECM remodeling could further impair mechanical integrity and force transmission, amplifying susceptibility to fasting-induced atrophy and contractile deficits...”

8. Methods - study design section: I suggest that this section be shortened and focused entirely on giving a broad overview of the main study design. I also suggest including a specific sub-section to further elaborate the development and characteristics of the different mouse strains investigated, along with a comment on the likely implications of these characteristics on the response to fasting.

Reply: We thank the reviewer for this suggestion. We had already revised and streamlined the Study design section in response to earlier peer review round, and we consider the current structure to provide a clear and coherent overview without the need for additional sub-sections. However, we agree with the reviewer that further clarification of strain characteristics and their potential implications is valuable. We have therefore added the following information:

“These strains differ markedly in muscle size—C57BL/6J exhibit normal muscle mass, BEH+/- have moderately larger muscles, and BEH mice show markedly enlarged muscles due to myostatin deficiency—features that were expected to influence their susceptibility to fasting-induced catabolic stress.”

9. Page 6, Line 12-13: Some more detail on the missing muscle samples would be useful. I assume that at least 116 isolated muscles should theoretically have been available (58 animals in the experiment, with 2 muscles extracted from each one), so it is not clear why only 96 were available for testing. While I recognize that some extent of tissue loss or failure is the reality in studies like this, the extent is concerning, considering how small some of the final sample sizes for certain subgroups and outcomes were, e.g., it appears that only a single EDL muscle was tested for the 100th contraction of the fasted BEH group. The varying sample sizes described in Table 2 may impact statistical power and capacity to interpret these results, and as such I suggest that this could be highlighted as a limitation of the study.

Reply: We thank the reviewer for this observation. As noted, a portion of the muscles was lost due to normal tissue damage or technical failure, which is not uncommon in isolated muscle experiments. This was particularly evident in the BEH strain, where fewer EDL muscles survived the full 100-cycle eccentric protocol. To account for these dropouts, we employed a linear mixed model approach, which is robust for repeated-measures data with missing values and thereby limits the impact of sample attrition on later contraction cycles. Nonetheless, we agree that the reduced sample sizes in certain subgroups represent a limitation and have now acknowledged this explicitly in the revised Limitations and future directions section.

“...Moreover, not all isolated muscles survived the full experimental protocol, particularly in the BEH strain, leading to smaller sample sizes in some subgroups; although the use of linear mixed models mitigated the statistical impact of such dropouts, this can be considered a potential limitation when interpreting later contraction cycles...”

10. Discussion: Overall, some very interesting points were made throughout the discussion and I appreciate the authors clear attempts to be balanced, and when speculating on potential mechanisms, that they made this clear.

Reply: We thank the reviewer for the positive feedback.

11. Page 7, Line 55-57: Unless I missed it, this "trend" was not mentioned in the results and looking at the results in Table 2, I am not clear on where this came from. If the authors consider this to be an important point to make, they should elaborate on why that is, otherwise I suggest just removing this sentence.

Reply: We thank the reviewer for spotting this, that was a mistake or typo and not supported indeed by our results. We removed accordingly the sentence.

12. Page 8, Line 6-7: In what way do these results indicate that myostatin inhibition could counteract muscle wasting in a catabolic state, considering that there was no difference for these outcomes between the strains?

Reply: We thank the reviewer for another good point and observation. We agree with the point raised and to avoid misinterpretation, we have revised the text accordingly. The updated version now reads:

“...Previous studies have reported conflicting findings regarding the role of myostatin inhibition in catabolic states, with some suggesting a protective effect against muscle wasting (14,45), and others showing the opposite, such as greater muscle loss in myostatin-dysfunctional BEH mice compared to wild-type during 12 weeks of 30% caloric restriction (46). In our study, however, no differences were observed across strains or between muscle types in response to fasting, suggesting that the severity of complete fasting may override genotype-related variability in phenotypic adaptations (15,46)...”

13. Page 8, Line 39 - 40: Considering that cells store only limited ATP regardless of feeding state, the key issue during fasting may relate to a reduced rate of ATP turnover due to limited substrate availability to sustain contractile function, rather than to low ATP availability.

Reply: We thank the reviewer another valuable observation. We have revised the text to reflect this point. The sentence now reads:

“This similarity suggests that fasting-induced reductions in contractile function might stem from shared mechanisms, such as a reduced rate of ATP turnover due to limited substrate availability or alterations in excitation-contraction coupling, which could equally impact both slow- and fast-twitch fibers (48,50).”

14. Page 10, Line 24 - 25: I appreciate the authors consideration of the limitations of their approach, but I also think that it is worth commenting on the benefits of examining isolated muscle preparations. I agree that this does not allow for consideration of full in vivo physiological and behavioral factors, but it also allows for isolation of peripheral mechanisms which I believe to be very valuable. Every methodological decision has a cost, and while I agree with the authors that it is useful to consider these costs, I also believe it useful to also comment on the strengths of the approach.

Reply: We thank the reviewer again for their consideration of our work and fully agree with the point raised. We decided to keep the text already highlighting the strengths of isolated muscle preparations (as justified in our reply to Reviewer 1) and have made a small update to further emphasize that this approach allows direct assessment of intrinsic muscle properties independent of systemic influences.

Second decision letter

MS ID#: bio.062245R1

MS Title: Metabolic Stress and Muscle Mechanics: Acute Response of Isolated Soleus and EDL Muscles to Prolonged Fasting in Mice with Distinct Muscle Phenotypes

Authors: Leonardo Cesanelli; Berta Ylaite; Marius Brazaitis; Nerijus Eimantas; Aivaras Ratkevicius; Danguole Satkunskiene; Petras Minderis

Dear Dr Cesanelli,

We have received some final comments from the one Referee to whom I returned the manuscript, and as you can see, they are happy with your edits and explanations. I am therefore happy to tell you that your manuscript has been accepted for publication in Biology Open, pending our standard publication integrity checks. It was accepted on 26th September 2025.